# Constraints to gene flow increase the risk of genome erosion in the Ngorongoro Crater lion population
Nicolas Dussex [1] ✉, Ingela Jansson [2], Tom van der Valk[3,4], Craig Packer [5], Anita Norman[2], Bernard M. Kissui[6], Ernest E. Mjingo[7] & Göran Spong [2,8] ✉

Small, isolated populations are at greater risk of genome erosion than larger populations. Successful conservation efforts may lead to demographic recovery and mitigate the negative genetic effects of bottlenecks. However, constrained gene flow can hamper genomic recovery. Here, we use population genomic analyses and forward simulations to assess the genomic impacts of near extinction in the isolated Ngorongoro Crater lion (*Panthera leo*) sub-population. We show that 200 years of quasi-isolation and the recent epizootic in 1962 resulted in a two-fold increase in inbreeding and an excess in the frequency of highly deleterious mutations relative to other populations of the Greater Serengeti. There was little evidence for purging of genetic load. Furthermore, forward simulations indicate that higher gene flow from outside of the Crater is needed to prevent future genomic erosion in the population, with a minimum of one to five effective male migrants per decade required to reduce the risk of long-term inbreeding depression and reduction in genetic diversity. Our results suggest that in spite of a rapid post-epizootic demographic recovery since the 1970s, continued isolation of the population driven by habitat fragmentation and potentially male territoriality, exacerbate the effects of genome erosion.

Increasing anthropogenic activities over the past centuries have had a strong impact on ecosystem health and diversity and are often characterised by severe wildlife population declines. Such declines have negative genetic consequences referred to as genome erosion (i.e., loss of genetic variation, increase in genetic load, mismatch between adaptations and environment[1]) which can increase the risk of extinction by trapping species into an extinction vortex[2].

The rate of demographic decline and recovery as well as duration of a bottleneck will have a strong influence on the magnitude of genome erosion[1,3]. For instance, only a small portion of genome-wide diversity may be lost if the decline is followed by quick demographic rebound, whereas a sustained bottleneck will likely result in a significant loss of diversity. Similarly, the amount of genetic load may not substantially change and thus impact individual fitness in future generations if there is a quick population recovery. In contrast, long-term declines characterised by gradual increases in inbreeding may facilitate a reduction in deleterious variation through purifying selection, in a process referred to as purging, whereas rapid declines may induce an increase in the frequency of deleterious variants[3,4], although both processes can occur in either types of declines due to stochastic effects.

There has been a strong focus on examining the effect of the intensity of population bottlenecks or founder events on genome erosion processes (e.g. ref. 5–9). However, even after a population recovers or stabilizes from such events, genome erosion can continue if the population remains isolated[1,3]. Habitat fragmentation associated with human activities reduces gene flow among subpopulations, increases genetic drift and inbreeding, and impacts genetic load[10]. High territoriality and competition from local individuals could also limit mating opportunities of immigrants and exacerbate this isolation[11]. For instance, in polygamous species where only few males contribute genetic diversity to future generations, genomic recovery (e.g.

[1]Department of Population Analysis and Monitoring, Swedish Museum of Natural History, SE-106 91 Stockholm, Sweden. [2]Molecular Ecology Group, Department of Wildlife, Fish, and Environmental Studies, Swedish University of Agricultural Sciences, SE-901 83 Umeå, Sweden. [3]Centre for Palaeogenetics, Svante Arrhenius väg 20C, SE-106 91 Stockholm, Sweden. [4]Department of Bioinformatics and Genetics, Swedish Museum of Natural History, SE-106 91 Stockholm, Sweden. [5]Department of Ecology, Evolution and Behavior, University of Minnesota, MN 55108 St. Paul, MN, USA. [6]School for Field Studies, Centre for Wildlife Management Studies, Karatu, Tanzania. [7]Tanzania Wildlife Research Institute (TAWIRI), Arusha, Tanzania. [8]Luke, FI 00790 Helsinki, Finland.
✉e-mail: nicolas.dussex@gmail.com; Goran.Spong@slu.se

https://doi.org/10.1038/s42003-025-07986-0  **Article**

reduction in inbreeding) can be relatively slow, and the random effect of genetic drift (i.e., loss of diversity or fixation of deleterious alleles) can be particularly strong as shown in the lek-breeding kākāpō (*Strigops habroptilus*)[6].

As a case in point, the Ngorongoro Crater (hereafter referred to as the Crater) lion population in Tanzania, which now comprises ~60 lions (~5 prides), seems to function as an ecologically isolated population. The Crater offers high prey abundance and minimal human threats whereas the surrounding landscape is a multi-use area shared with pastoralists and their livestock, where natural prey densities are lower and more variable, and where human-lion conflicts occur. Thus, opportunities for pride establishment outside the Crater are limited[12–14]. Importantly, even though there are no geographical barriers to dispersal into the Crater[14], immigration of breeding males from the adjacent Serengeti plains has been infrequent, with only one breeding male successfully establishing in the Crater between 1965 and 2013[13,15]. Furthermore, intensive monitoring through individual identification indicates a high degree of territoriality of local resident males and little opportunities for immigrant males to mate and establish[13]. For instance, 80% of resident males were born in the Crater, whereas in the rest of the Serengeti National Park only 33% of resident males were born locally. Moreover, a higher proportion (i.e., 7 vs <1%) of males became resident in their natal pride in the crater compared to the in Serengeti. Lastly, tenure is more successfully retained in the Crater, with males siring nearly ten times as many cubs in the Crater relative to the rest of the Serengeti[13]. Consequently, reduced reproduction opportunities of immigrant males from outside of the Crater cause reduced gene flow and to the genetic isolation of the population. This may be counteracted by inbreeding avoidance, as it has been found that mating among related pride members are rare[16,17].

Consistent with an estimated 85% reduction in the range of African lions since 1500 AD[18] primarily driven by increasing anthropogenic pressure on the landscape, the Crater population experienced several bottlenecks associated with landscape fragmentation. Exponential human population growth and land use changes (i.e., agriculture and pastoralism[19]), retaliatory and ritual lion hunts[20,21], colonial-era trophy hunting[22] as well as periodic rinderpest epizootics between 1890-1962 reducing ungulate populations in the Greater Serengeti Ecosystem (GSE) exacerbated this population fragmentation over the past ~200 years. Moreover, the population experienced a severe decline in 1962 induced by an epizootic outbreak causing pronounced lethargy and skin lesions followed by death, after which 15 adults and juvenile survivors founded the current population[12,13]. After a period of recovery, the population declined in the late 90 s for unknown causes[23] and again in 2001, from a combination of tick-borne disease and canine distemper virus (CDV)[24], with the number of lions stabilising at 50 individuals over the past ~30 years. In spite of the immigration of a few breeding males in 1964–1965 and between 2013 and 2018, the Crater population is still characterised by low diversity and inbreeding depression. For instance, there is evidence for increase in levels of sperm abnormality and low spermiogenesis[12,13,25–27] as well as higher levels of cub mortality in the first few weeks of life[13]. These observations seem consistent with a population harboring a significant genetic load. However, little is known about the impact of population declines and isolation on genome-wide variation of Crater lions and about the distribution of the deleterious variation in the population.

When temporal data are not available, comparative approaches can be useful for genomic studies in endangered species as they can reveal contrasted patterns of genome erosion, reflecting distinct population histories or degrees of connectivity (e.g., Ibex[28]; Svalbard reindeer[29]; Indian tigers[30]). Here, we analyse 15 newly-sequenced lion genomes from Tanzania (the wider Ngorongoro Conservation Area including the Crater and the adjacent Serengeti plains) and compare them to 5 published lion genomes from Tanzania, Botswana and South Africa to examine the genomic consequences of the recent population decline and isolation on the Crater population. Genomic data produced in this study suggest that the recent decline induced an increase in genome erosion and realised load in the Crater population while simulations suggest that reduced male immigration could have also potentially contributed to the observed pattern. Simulations

also reveal that a minimum of one to five effective male migrants per decade would be required to avoid substantial reduction in genome-wide variation and increases in inbreeding and genetic load.

## Results and Discussion

### Population structure, gene flow and past demography

We analysed 20 lion genomes to reconstruct the population history of the Crater lions and to examine the consequences of isolation and recent population decline on genome-wide diversity and genetic load of the population (Fig. 1a).

We first examined the population structure to determine genetic relationships among African lion populations and assess whether the Crater population is distinct from other African populations. Overall, there was little evidence for admixture with adjacent Serengeti plains lions (Fig. 1b, c). While there was strong support (i.e., lowest Cross Validation error) for two genetic clusters (K = 2), considering three clusters (K = 3) supported a clear division among Crater, Greater Serengeti (i.e., comprising Tanzania, Ndutu, Endulen) and Selous/Botswana/South African lions (Fig. 1b, c). We also note that two males of unknown natal origin (Cra_05, Cra_07) and one Crater-born female (Cra_06; Supplementary Data 1) show slightly distinct ancestry.

Estimates of the divergence time between the Crater and Greater Serengeti suggests that gene flow was reduced c. 200 y. BP (Mean: 178; 95% HPD: 41- 380; Supplementary Fig. 1). The timing of this divergence is consistent with the recent bottleneck and increase in habitat fragmentation of the past 200 y. BP which both contribute to the genetic isolation of the population.

The demographic reconstruction based on the Pairwise Sequentially Markovian Coalescent (PSMC) showed a long-term decline over the past ~1 Ma BP and shared demographic history among lion populations (Fig. 2a), as previously shown by de Manuel et al. [31] for other lion populations. This decline is consistent with drying and cooling conditions of the first 1 M of the Pleistocene, while the Eemian period (c. 130,000 to 115,000 y. BP) characterised by warmer and wetter conditions and expansion of savannas and forests showed a plateau (Fig. 2a). Subsequently this decline continued with the next cycle of cooling[32] until the Holocene.

The recent past demography (i.e., ~200 generations or ~1000 y. BP[33]) indicated a population increase c. 1200-1000 y. BP, coinciding with a prolonged dry period in Eastern and Equatorial Africa[22], followed by a decline c. 600 y. BP, coinciding with wetter conditions of the Little Ice Age[34] (Fig. 2b). While speculative, this suggests that denser woodland may have been potentially less favourable to lions. Population census size estimates since the early 1960s reflect the effects of the 1962 epizootic and the decline of 2001 caused by CDV, with population size ranging between 10 and 124 individuals (Fig. 2c).

### Genetic diversity and load

Heterozygosity and inbreeding estimates revealed that the Crater population has the lowest amount of variation and is ~1.6 times as inbred compared to the larger neighbouring Serengeti and Selous populations ($F_{ROH-Crater} = 0.37 \pm 0.031$, $F_{ROH-Serengeti} = 0.22 \pm 0.020$, $F_{ROH-Selous} = 0.23 \pm 0.029$; Fig. 3, S2; Supplementary Data 4). Furthermore, there is on average ~60% (range: 51-66%) of the $F_{ROH}$ coefficient comprising Runs of Homozygosity (ROH) $\geq$ 2Mb, consistent with recent inbreeding events. When using a recombination rate estimated in felids[35] and a generation time of 5 years for lions[31], inbreeding events characterised by ROH $\geq$ 2Mb and ROH $\geq$ 10 Mb date to the past ~110 and ~30 years, respectively (Supplementary Data 5). These signatures of inbreeding are thus consistent with the recent history of population decline resulting from habitat fragmentation over the past two centuries in the region (Fig. 2c).

$F_{ROH}$ estimates for the newly-sequenced Serengeti and Selous lions in our dataset are consistent with previous estimates for Tanzanian lions ($F_{ROH} \simeq 0.2$; Supplementary Data 4)[31].

We estimated genetic load by annotating variants in coding regions using Snpeff[36]. When considering the frequency of deleterious variation

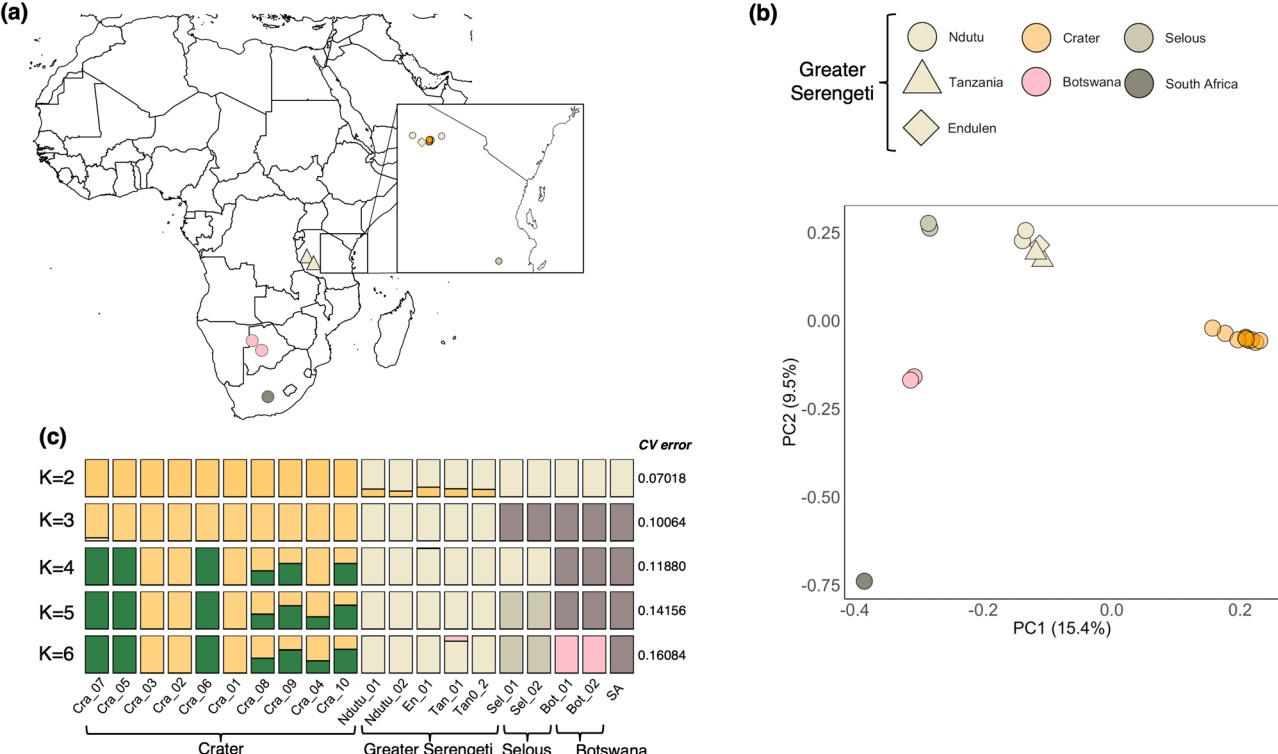

**Fig. 1 | Sampling, population structure and gene flow. a** Sampling locations for African lions. Inset shows the 15 newly-sequenced genomes for this study including 10 Ngorongoro Crater, 3 Greater Serengeti and 2 Selous* lions. **b** Principal Component Analysis. **(c)** Admixture plot for K = 2-6 and for the 15 newly-sequenced and five lion genomes from de Manuel et al. [31]. Cross Validation error values are given. *Note that Selous is nowadays known as Nyerere National Park.

among populations, the $R_{xy}$ ratio[37] showed an excess in High impact (i.e., premature stop codons) variants and a slight deficit for Moderate impact (i.e., non-disruptive variants that might change protein effectiveness) variants in Crater lions relative to Serengeti and Selous lions (Fig. 4a). While the total load estimate for High and Moderate impact variants was not significantly different among populations, the High impact load was higher in the Crater population relative to other populations (Fig. 4b; Supplementary Data 4).

Since most deleterious mutations are partially recessive and since heterozygous ones are hidden from selection, homozygous mutations are the most informative of the negative fitness effects, also referred to as the realised load[1,3]. Consistent with the higher inbreeding in the Crater population, we found significantly higher realised load for both High and Moderate impact variants relative to the more outbred populations (i.e., Serengeti and Selous; Figs. 4c, S3). Furthermore, the numbers of heterozygous variants for both High and Moderate categories are significantly lower in Crater lions relative to the Serengeti population (Supplementary Fig. 3), suggesting that those may have been lost through a combination genetic drift directly associated with the recent bottlenecks or through some early purging effect[1,3]. Nevertheless, the higher total load in several Crater lions relative to other populations (i.e., Serengeti, Selous and South Africa) suggests that there has not been sufficient time for selection to purge highly deleterious variation and that the Crater population is most likely in the early stages of exposure to genome erosion following the ~200 years isolation from the Serengeti population and the 1962 epizootic.

As theory, empirical data and simulations indicate, the early stage of a decline is characterised by an increase in inbreeding and realised load and thus of inbreeding depression[3,4]. Lethal or highly deleterious alleles should be reduced in frequency relatively early during a bottleneck, while a number of moderate impact variants with a lower selection coefficient and thus less individual effect on fitness are more likely to drift to fixation during a bottleneck and still contribute to inbreeding depression[3]. The pattern observed in Crater lions is similar to that of Grauer's gorilla (*Gorilla beringei*

*graueri*), which experienced severe population declines over the past century[38]. However, the random effect of drift is particularly evident in Crater lions, with mostly High impact variants drifting to high frequencies, whereas a portion of the Moderate impact variants may have been lost during the bottleneck. Alternatively, it is possible that the Serengeti and Selous lions lost more of their High impact variants through a combination of purging or drift, which would cause the same observed difference. Finally, we note that the South African genome (i.e. previously identified as genetically grouping with South African lions[31]) and most likely from a zoo, shows one of the lowest total load, consistent with a scenario of purging, but also has high inbreeding and realised load.

Based on prior evidence of inbreeding depression, sperm abnormality and low sperm counts in Crater lions[12,25–27], as well as higher cub mortality in the first weeks of life[13], we examined the gene functions of genes carrying High and Moderate impact alleles (Supplementary Data 6) using the Mouse Genome Informatics database[39]. Among genes carrying High impact variants (i.e., premature stop codons) at higher frequency in the Crater population relative to Serengeti and Selous lions, we found a number of genes associated with sperm morphology and male fertility (e.g., *AURKC, TSHB, TLR6, CWF19L2*), nervous system (e.g., *ARSG, FGD4*) and immunity (e.g., *TLR6*; Supplementary Data 7). Among genes carrying Moderate impact variants (i.e., Missense) at high frequency, we again found functions associated with sperm morphology and male fertility (e.g., *NUTM1, CCDC87, CEP250, PRSS55, TSPYL1*) and additionally, with cardiovascular system (e.g., *LAMA4, KLK5, GAS2L3, PPP1R15B, OBSCN*) and nervous system (e.g., *LAMA1, KCNJ16, SCN1B*; Supplementary Data 7).

### Forward genome-informed simulations

To model the evolutionary trajectory of populations and to assess the plausibility of interpretation based on empirical data[40], we performed forward-in-time genomic simulations recapitulating the population history of Crater lions based on our demographic reconstructions and population monitoring data since the 1970s using SLiM 4.0[41,42] (Supplementary Data 8).

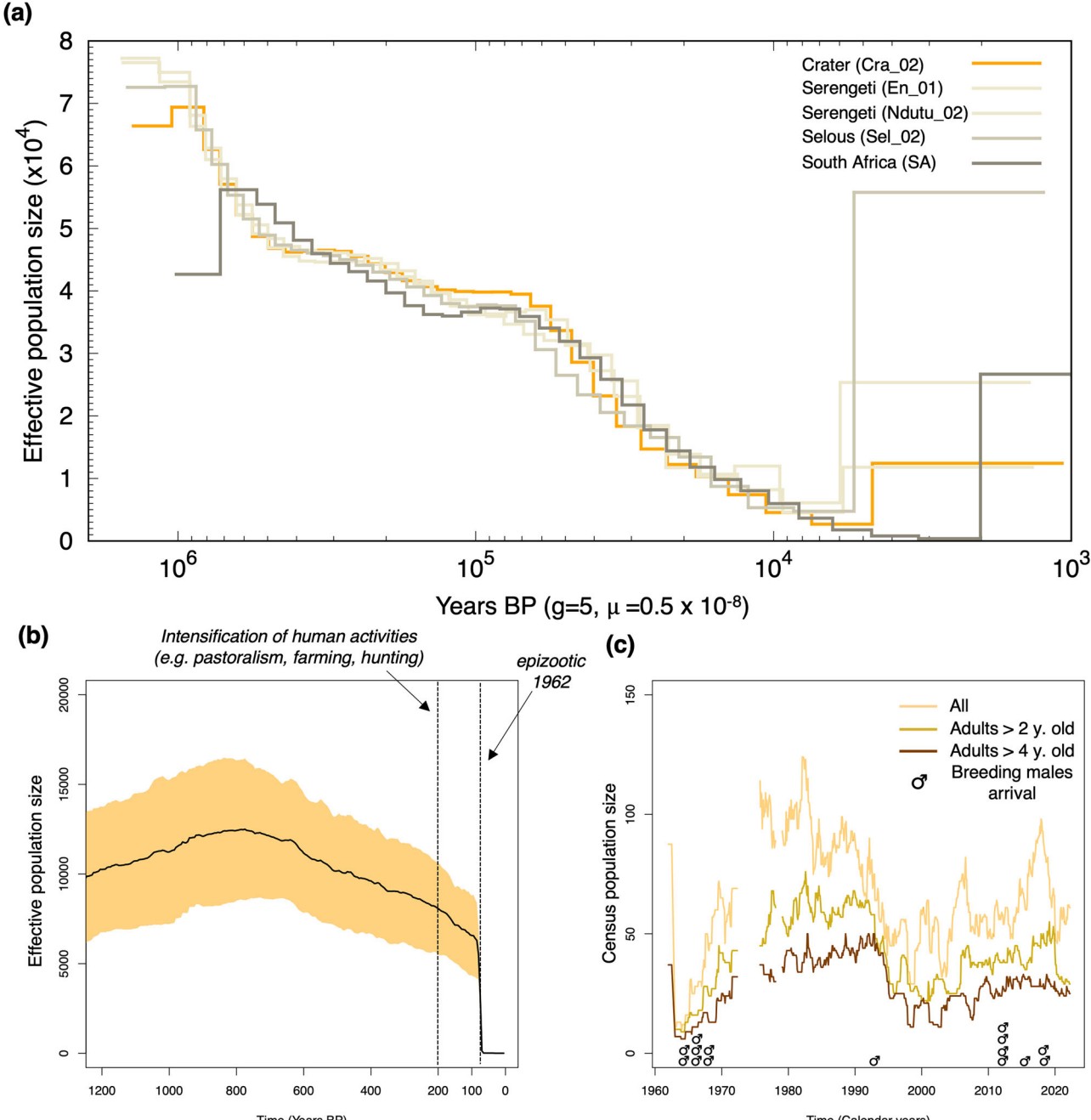

**Fig. 2 | Past demography. a** Long-term past demography using the PSMC with a substitution rate of $4.5 \times 10^{-9}$ substitutions/site/generation and a generation time of 5 years[31]. **b** Recent past demography for Crater lions over the past ~200 generations and assuming a generation time of 5 years[31]. The black line depicts the mean and shaded area the 95% CI for 10 replicates (Supplementary Data 2) each calculated as the geometric mean over 40 independent estimates from the observed spectrum of linkage disequilibrium using the *Panthera* sp. recombination rate[35]. **c** Population census size from 1962 to 2022. Note that no data was collected between 1972-75 and 1978 and 1980 (Supplementary Data 3).

Overall, our simulations were consistent with our empirical data (Fig. 5). We found a 2-fold increase in inbreeding (from $F_{ROH}$ 0.14 to 0.27) and ~15% reduction in heterozygosity in the Crater population relative to the Greater Serengeti Ecosystem (GSE) over the past 200 years (Fig. 5a). This increase was gradual since the population split c. 200 y. BP. However, while the population was reduced to 9 females and one male after the 1962 epizootic, the immigration of reproducing males right after the bottleneck and rapid demographic recovery (Fig. 2c) induced a rapid ~20% reduction (from $F_{ROH} \sim 0.25$ to 0.2) in inbreeding until the 1970s when inbreeding started to gradually increase.

There was also a ~ 50% increase in realised load relative to the GSE population (Fig. 5a), consistent with increasing inbreeding and with the estimated realised load based on empirical data (Fig. 4c). Moreover, the masked load showed a temporal decrease of <1% over 200 years as it is gradually converted to realised load via the effects of inbreeding and drift since the time of the isolation of the Crater population. This decrease is consistent with the lower number of heterozygous High impact variants relative to GSE (Supplementary Fig. 3). However, the migration of seven males in 1964-1965 and between 2013 and 2018, led to a reduction in inbreeding and realised load and induced an increase in heterozygosity and masked load.

**(a)**

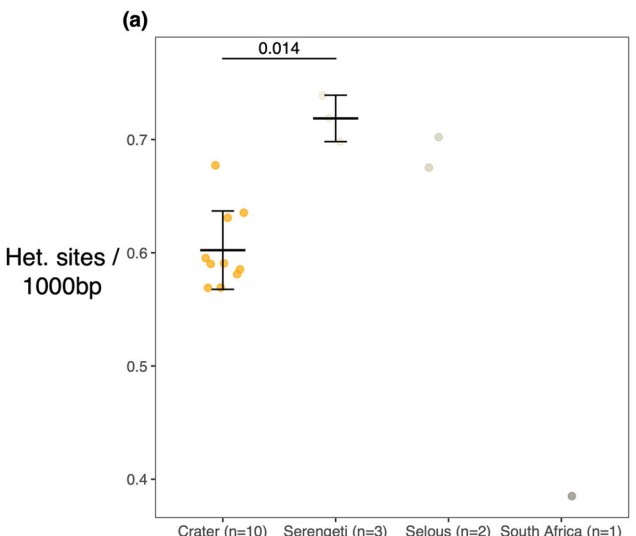

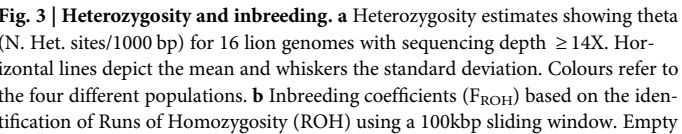

**(b)**

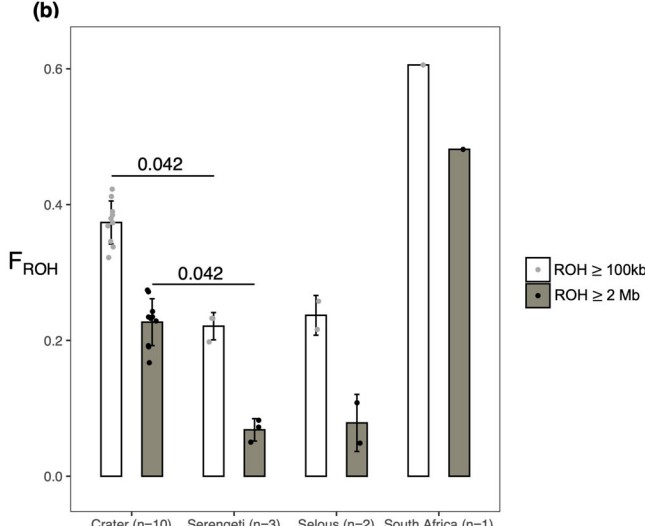

**Fig. 3 | Heterozygosity and inbreeding. a** Heterozygosity estimates showing theta (N. Het. sites/1000 bp) for 16 lion genomes with sequencing depth ≥ 14X. Horizontal lines depict the mean and whiskers the standard deviation. Colours refer to the four different populations. **b** Inbreeding coefficients (F$_{ROH}$) based on the identification of Runs of Homozygosity (ROH) using a 100kbp sliding window. Empty bars show the proportion of genomes in ROH ≥ 100 kb (i.e., background relatedness) and grey bars the proportions in ROH ≥ 2Mb (i.e., recent inbreeding events). Bars extending from the mean values represent the standard deviation. Statistical significance was assessed with Wilcoxon signed-rank tests ($n = 4$). Only significant differences are shown.

In contrast, the number of deleterious variants in any category did not vary substantially between the Crater and GSE. Nevertheless, there was an indication of reduction in Very Strongly deleterious variants resulting from the isolation of the population c. 200 y. BP (Fig. 5b). This is not entirely surprising since the most deleterious variants should be purged relatively early during a bottleneck through purifying selection as they have the most impact on fitness[3]. Furthermore, a small number of simulations showed increases in those variants following the epizootic of 1962 and the migration of seven males in 1964-1965 (Fig. 5b).

We ran our simulations over an additional 100 years period, until the year 2120 to assess the effect of male migration (i.e., 0 to 10 effective migrants per decade) on genome-wide variation. These simulations indicated that between one to five effective male migrants per decade would be required to remain within a 5% window of change in heterozygosity, inbreeding and load (Fig. 6) for a realistic carrying capacity K of 50 to 100 individuals for the Crater population. Below one effective migrant per decade, there would be a substantial risk of negative genetic effects, with >40% and >20% increase in inbreeding and realised load, respectively. There would also be a > 10% reduction in heterozygosity, especially for a K of 50 individuals. Consistent with theory[3] and empirical data[43], low migration (i.e., M = 0-1) would induce a reduction in masked load whereas higher migration (i.e., M > 1) would lead to an increase through the introduction of new genetic variation. These effects would be stronger for smaller K values, where overall genetic diversity is likely to be lower than for larger K values. Thus, with little to no migration over 100 years, the masked load would be reduced through the effect of purging, whereas overall diversity would also be reduced (Fig. 6d).

**Conservation implications**

In spite of severe population declines driven by landscape fragmentation and an epizootic, the Ngorongoro Crater lion population has recovered quickly since the 1970s. Yet, the population shows evidence of inbreeding and inbreeding depression. Our simulations show that rapid demographic recovery and periodic effective migration can counteract the negative genetic effects of these bottlenecks and our empirical data provide an example of how continued geographical isolation driven by habitat fragmentation may negate the positive effects of population recovery by exacerbating genetic drift. While speculative, this drift effect may be affected by male territoriality that reduces the opportunities for immigration and the establishment of breeding status of males from outside the Crater, thereby

exacerbating the effects of genome erosion. Taken together, our results supports theory, which posits that demographic increase and stability alone is not a panacea for preventing loss of genetic variation and that gene flow is required to enable the genomic recovery of small populations[44,45].

Our simulations and a growing number of empirical studies[6,28,30,38] show that a gradual increase in inbreeding could facilitate the reduction of some of the genetic load through purging. Thus, while gene flow from an outbred to an inbred population will contribute to a genetic rescue effect[44,45], it will also represent a risk of introducing deleterious variation. In a highly inbred population, this newly introduced deleterious variation could readily be expressed, reducing overall fitness and thus increasing the risk of population extinction. For instance, the inbred Isle Royale wolf[46] population experienced a rapid population decline after the effective migration of a single male. Consequently, fostering periodic and long-term gene flow into the Crater before the population becomes too inbred would reduce the risk of expression of deleterious variation and population collapse.

Given the logistical challenges of translocations and the possible behaviourally-mediated resistance of immigration into the small Crater population[13], fostering long-term connectivity with the GSE would increase the chance of effective migration (i.e., mating) events or the temporary interactions of dispersing Serengeti males just outside the Crater with Crater-born females[15,47]. Mitigation of human-lion conflicts in the connecting landscape between the Crater and Serengeti to maintain, and possibly further improve, dispersal opportunities[15] thus likely represents the best long-term management option to reduce the risk of future genome erosion in Crater lions.

Our study improves our understanding of genome erosion in fragmented populations driven by human activities. Importantly, combining empirical genomic data with forward simulations provides deeper insights into the dynamics of genetic variation and genetic load[40] and of the threats of genome erosion in endangered populations.

**Methods**

**Ethical statement**

We have complied with all relevant ethical regulations for animal use. All research fieldwork and data collection were conducted in accordance with the Tanzania Wildlife Research Institute's regulations. It was carried out under the yearly renewed research permits granted to our research project titled "Balancing Pastoralist Livelihoods and Wildlife Management in

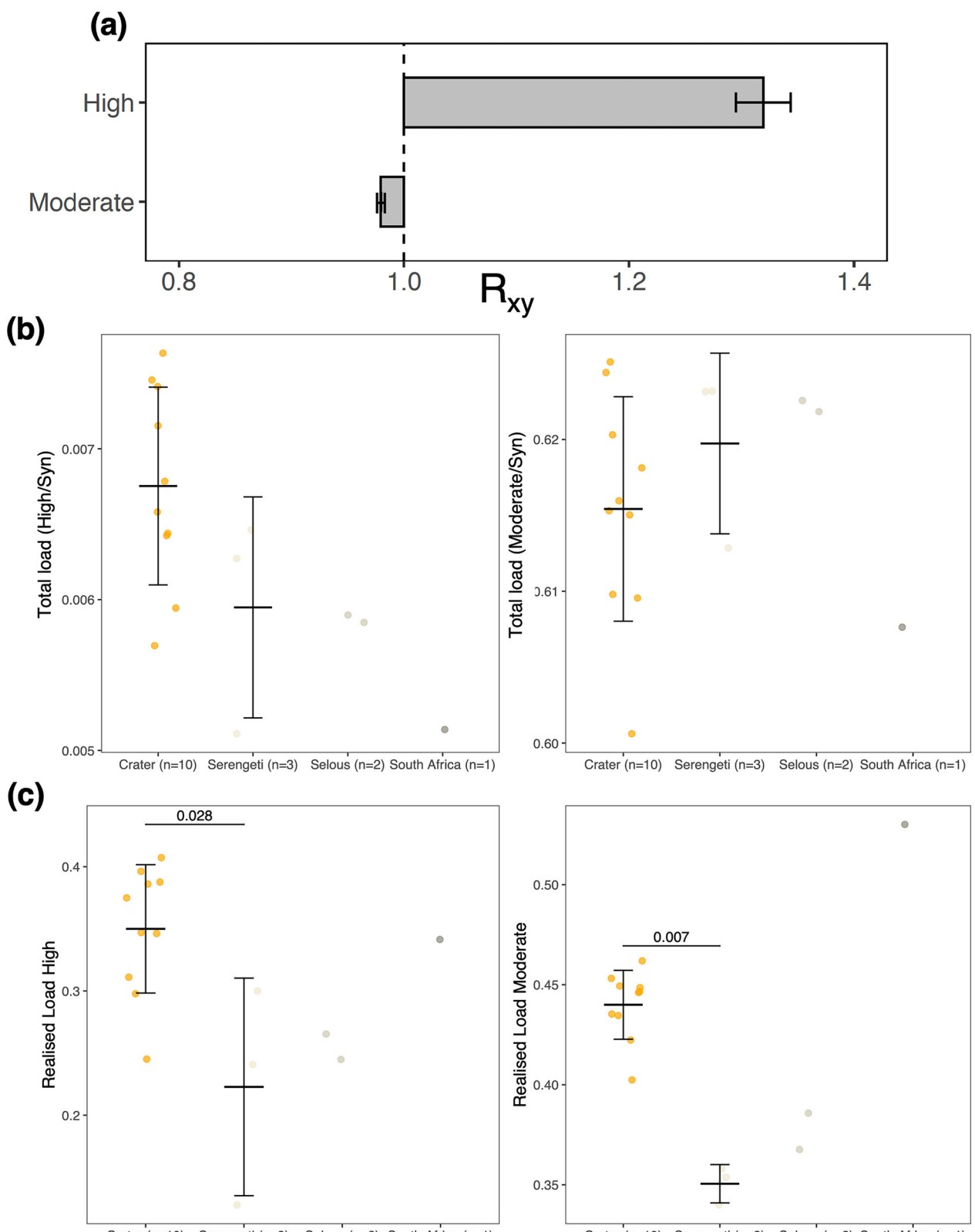

**Fig. 4 | Genetic load. a** $R_{xy}$ of derived alleles for high- and moderate-impact variants for Crater lions relative to Serengeti and Selous lions. $R_{xy} < 1$ or >1 corresponds to a deficit or an excess in allele frequency, respectively, in population x (i.e., Crater, $n = 10$) relative to population y (Serengeti and Selous lions, $n = 5$). Whiskers represent ±1 SD. **b** Total load estimated as the ratio of High and Moderate impact to Synonymous variants for 16 lions (with sequencing depth ≥ 14X). **c** Realised load. Horizontal lines depict the mean and whiskers the standard deviation. Statistical significance was assessed with Wilcoxon signed-rank tests ($n = 4$). Only significant differences are shown. Colours refer to the four different populations.

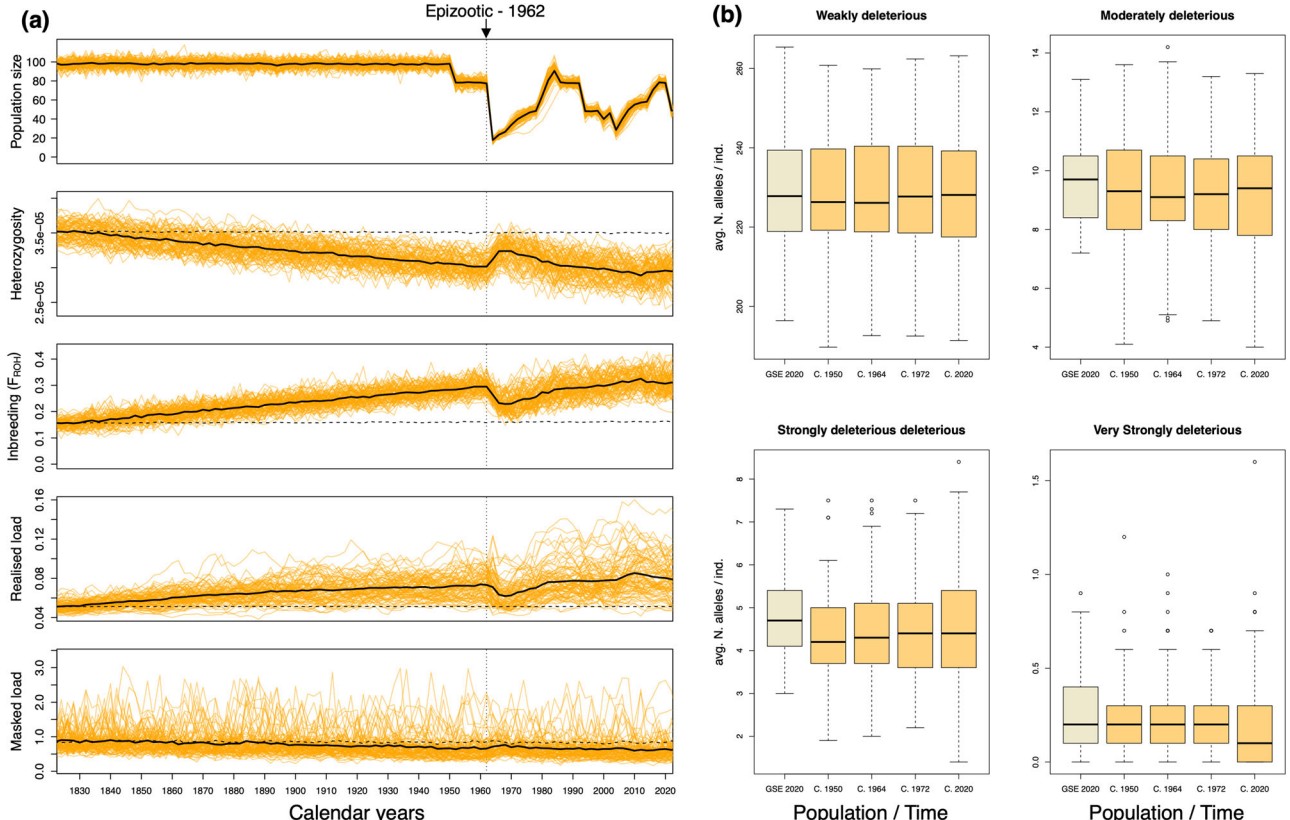

**Fig. 5 | Genome-informed simulations recapitulating the recent population history of Crater Lion from 1820 to 2020. a** Temporal changes in census size (N), mean heterozygosity, mean inbreeding ($F_{ROH}$), mean realised load and masked load (rescaled for 20,000 genes; See Methods). Dashed and full lines depict the mean across all runs for the Greater Serengeti Ecosystem (GSE) and Crater (C.) populations, respectively. **b** Average number of deleterious mutations of each category (for 5000 genes) for the GSE (2020) and the Crater population (1950, 1964, 1972 and 2020). Note that 1964 corresponds to 2-years and 1972 corresponds to 10 years (i.e. 2 generations) after the epizootic, respectively. Horizontal lines within boxplots depict the median, bounds of boxes represent the first and third quartiles and whiskers extend to 1.5 times the interquartile range.

Ngorongoro", and to each individual researcher, by the Tanzania Commission for Science and Technology (COSTECH; Dar es Salaam, Tanzania; rclearance@costech.or.tz) and Tanzania Wildlife Research Institute (TAWIRI; Arusha, Tanzania; researchclearance@tawiri.or.tz).

## Data collection and sequencing

We obtained tissue samples for 15 wild lions from the Ngorongoro Conservation area (NCA; n = 13), and Selous (n = 2; Supplementary Data 1). The NCA lion samples were collected in the Crater (n = 10), Ndutu (n = 2; Serengeti plains, on border of Serengeti National park), and Endulen (n = 1; between the Crater and Serengeti, from a lion born in Ndutu). Samples from lion were collected using biopsy darting, or by extracting a small piece of tissue from the ear using a biopsy punch on animals immobilised for other reasons, e.g. collaring, and from a dead lion. Samples selected for the WGS were selected from the more distantly related individuals (i.e., different prides, different parents), based on our observation data. Relatedness was estimated with vcftools[48] using the –relatedness flag and ranged between 0.09 and 0.16, indicating distant relationships among genomes. DNA extraction was performed using a DNeasy Blood & Tissue Kit (Qiagen, Hilden, Germany) on a Qiagen Symphony extraction robot. Genomic library preparation from modern DNA extracts was performed using a Illumina TruSeq DNA PCR-free Library Preparation Kit. Paired-end libraries were built with 150 bp size selection using 1000 ng sheared DNA input, at the Science for Life Laboratories (SciLifeLab), Stockholm, aiming for a minimum target coverage of ~15X. Libraries were sequenced using one Illumina HiSeq X lane using a 2x150bp setup.

Additionally, we obtained genomic data for 5 lions from Tanzania (n = 2), Botswana (n = 2) and South Africa (n = 1; PRJNA611920[31]). Based on the PCA clustering results, we grouped our samples for downstream analyses such as: Crater (n = 10), Serengeti (n = 5), Selous (n = 2), Botswana (n = 2), South Africa (n = 1).

## Genome data mapping

We processed raw genomic data using the GenErode bioinformatics pipeline[49]. Adapter trimming was done with fastp v0.22.0[50] and reads were mapped to the chromosome-level Lion assembly (P.leo_Ple1_pat1.1; GCF_018350215.1) using BWA-MEM v0.7.17[51]. Read were then sorted using SAMtools v1.12[52], duplicates removed with picard MarkDuplicates v2.26.6 (http://broadinstitute.github.io/picard/), and reads realigned around indels using GATK IndelRealigner v3.4.0[53].

We called variant using the mpileup command of bcftools v1.8[54], filtering out variants using a minimum depth of coverage (DP4) of ~1/3 (i.e., 5X) of the average depth of coverage, and base quality QV ≥ 30. Indels and SNPs within 5 bp of indels were removed. We also filtered out SNPs in heterozygous state that were not in an allelic balance (i.e., number of reads displaying the reference allele/depth) of < 0.2 and > 0.8 in order to avoid biases caused by contamination, mapping or sequencing error. After merging all individual vcf files, we excluded the sex chromosomes (NC_056697.1, NC_028302.1) and masked repeats with BEDtools v2.27.1[55].

We retained 20 genomes for the Principal Component Analysis (PCA) without filtering for missing data and obtained 4,560,409 SNPs. For estimates of heterozygosity, inbreeding and genetic load, we only retained the 16 genomes with the highest coverage (i.e., ≥ 14X; Supplementary Data 1). After filtering for missing data with bcftools we retained 3,308,190 SNPs.

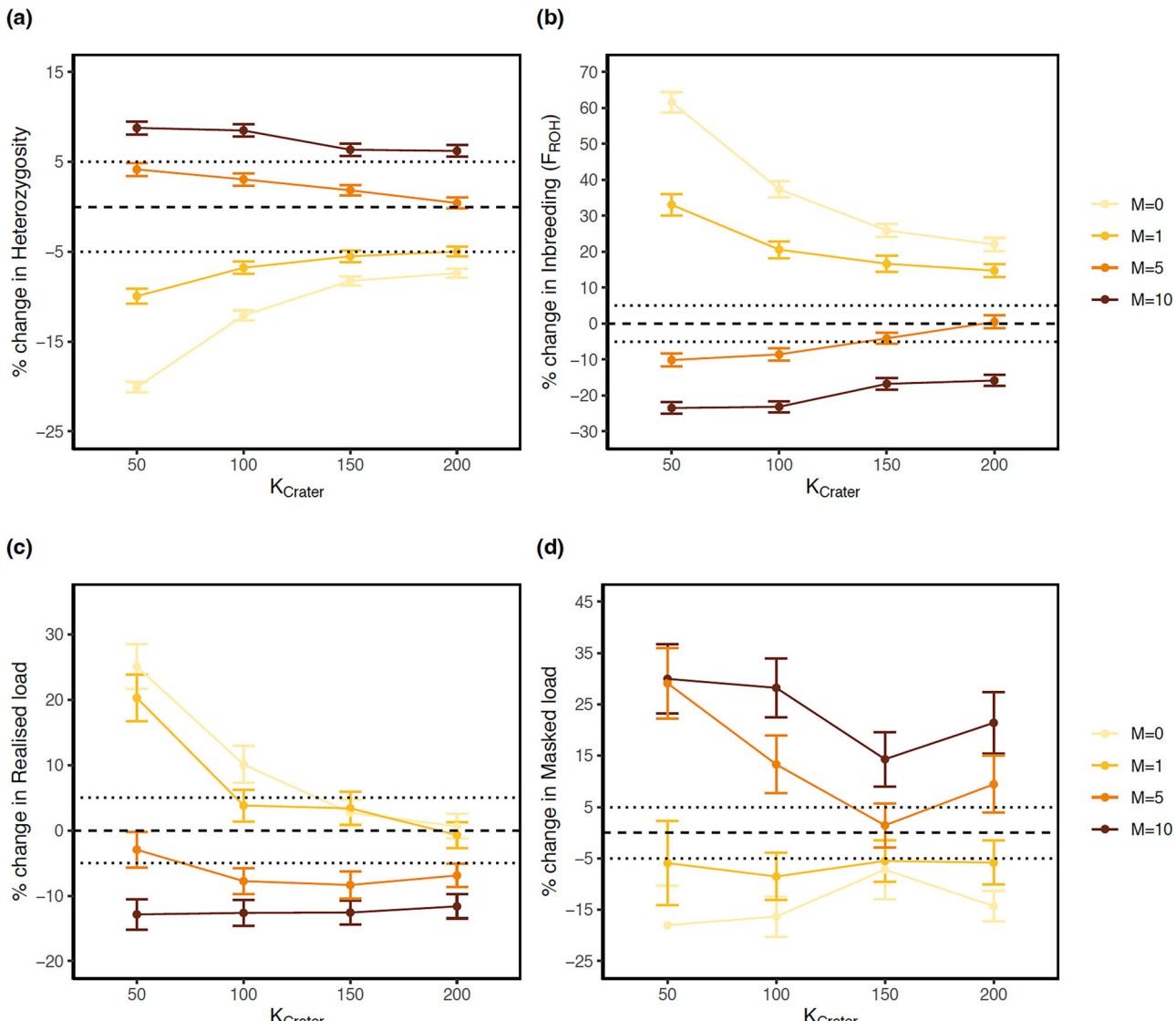

**Fig. 6 | Prediction of future changes in genome-wide variation over 100 years for various migration rates. a** Heterozygosity; (**b**) Inbreeding $F_{ROH}$; (**c**) Realised load; (**d**) Masked load. We simulated 0, 1, 5 and 10 effective migrants per decade and $K_{Crater}$ values of 50 to 200. Values > 0 and <0 indicate increase and reduction, respectively. Points represent mean and whiskers represent the 95% CI. Dotted lines depict a 5% change and dashed line no change.

## Population structure and past demography

We first performed a PCA in PLINK v2[56]. We then estimated individual-based ancestry to infer the number of genetic clusters with ADMIXTURE v1.3.0[57] for K = 1–6 and using the cross-validation error estimation (--cv option). We estimated past fluctuations in effective size ($N_e$) using two complementary approaches. First, we used the Pairwise Sequentially Markovian Coalescent (PSMC) v0.6.5[58]. We excluded the X chromosome and generated consensus autosomal sequences for five high-coverage genomes with SAMtools mpileup[52], using base and mapping quality filters (-Q 30, -q 30) and with the vcf2fq command from vcfutils.pl. We excluded sites with depth < 5X and excluded positions with more than two times the average coverage estimated for each genome (i.e. ~30-40X). To infer the TMRCA between each chromosome from each individual genome, we used the following parameters: Number of iterations, N = 25; Tmax, t = 15; atomic time interval, p = 64 (4 + 25*2 + 4 + 6, for each of which parameters are estimated with 28 free interval parameters). We used a substitution rate of $4.5e^{-9}$ substitutions/site/generation and a generation time of five years[31]. Secondly, we reconstructed the past demography of Crater lions over the past 200 generations using GONE[33] which estimates changes in $N_e$ calculated as the geometric mean over 40 independent estimates from the

observed spectrum of linkage disequilibrium (LD). We used autosomal chromosomes and the following parameters: PHASE = 2; cMMb=1.11[35]; DIST = 1; NGEN = 2000; NBIN = 400; MAF = 0.0; ZERO = 1; maxN-CHROM=85; maxNSP=50000; REPS = 40; threads = −99. We reduced the hc value from the default of 0.05 to 0.01 to avoid biases caused by recent immigration as suggested by Santiago et al.[33]. We performed 10 replicate runs for the population and assumed a generation time of five years[31].

We also estimated population divergence time between the Crater (n = 10) and Serengeti (Ndutu, n = 2; Endulen, n = 1; Tanzania, n = 2) for nuclear data using SMC + + v1.15.2[59]. We used a substitution rate of $4.5e^{-9}$ substitutions/site/generation[31] and used the 'estimation validation' approach with –em-iterations 5000, and –thinning 1300, –regularization-penalty 6, –polarization-error of 0.5, –ftol of 1e-7, –c 1000000 and –xtol of 1e-7. We reconstructed the SFS for each population separately and the joint SFS between two populations using the smc + + vcf2smc command. We then used the smc + + split command to calculate the marginal estimate of joint demography. For this analysis, we performed 50 bootstrap replicates for each population and for 15 chromosomes to reduce computational time and estimate the mean and 5th and 95th percentiles. We also assumed a generation time of five years[31].

## Heterozygosity and inbreeding

We first used mlRho v2.7[60] to estimate the individual mutation rate ($\theta$), which approximates the genome-wide heterozygosity measured as the number of heterozygous sites per 1,000 bp. We down-sampled each genome to the average coverage of the genome with lowest coverage (i.e., 14X), filtered out bases with quality (-Q) < 30, mapped sequencing reads with mapping quality (-q) <30 and positions with root-mean-squared mapping quality (MQ) < 30 from the historical and modern bam files. We also filtered out sites with depth <2X and higher across all our genomes to avoid false heterozygous sites.

We identified runs of homozygosity (ROH) using the sliding-window approach implemented in PLINK v2. We estimated inbreeding coefficients ($F_{ROH}$) by dividing the sum of all ROH by the size of the genome (autosomes only). We used the following parameters: a sliding window size of 100 SNPs (*homozyg-window-snp 100*); no more than 1 heterozygous site per window to assume a window as homozygous (*homozyg-window-het 1*); at least 5% of all windows including a given SNP to define the SNP as being in a homozygous segment (*homozyg-window-threshold 0.05*); a homozygous segment was defined as a ROH if the segment included ≥ 25 SNPs (*homozyg-snp 25*) and covered ≥ 100 kb (*homozyg-kb 100*); the minimum SNP density was one SNP per 50 kb (*homozyg-density 50*); and the maximum distance between two neighbouring SNPs was ≤ 1,000 kb (*homozyg-gap 1,000*). Finally, we set the value at 750 heterozygous sites within ROH (*homozyg-het 750*) in order to prevent sequencing errors to cut ROH. We statistically compared heterozygosity and $F_{ROH}$ among populations using Wilcoxon signed-rank tests in R[61] for ROH ≥ 100 kb (i.e., background relatedness) and ROH ≥ 2Mb (i.e., recent inbreeding events).

Using the length of ROH, we also inferred the timing of inbreeding by solving $g = 100/(2rL)$[62], where $g$ corresponds to the number of generations back in time, $L$ to the length of ROH in Mb, and $r$ to the recombination rate. We used a $r$ of 1.11 cM/Mb estimated in felids[35] and a generation time of 5 years[31]. Inferred times based on ROH lengths are shown in Supplementary Data 5.

## Genetic load

To estimate genetic load, we first generated an ancestral felid genome using domestic cat (SRR1179888-SRR1179901), tiger (SRR13242485) and cheetah (SRR22273180) to polarise our multi-individual vcf. We mapped short reads to the lion assembly (GCF_018350215.1) as described in the 'Genome data mapping' section and subsampled each of these three genomes to a depth of 6X and merged them with samtools merge. We then used ANGSD v0.917[63] to generate a consensus ancestral genome using the doFasta 2 and doCounts 1 options. Next, we used a custom script, to polarise the vcf to the ancestral allele (see Code availability section).

We then annotated synonymous and non-synonymous variants in coding regions using SnpEff v4.3[36]. We removed gene models with in-frame STOP codons, missing START and terminal STOP codons (-J option) and genes labelled as pseudogenes (--no-pseudo option) with Cufflinks v2.2.1[64] and obtained a total of 19,491 genes. We identified three categories of variants: a) Synonymous; b) Moderate: non-disruptive variants that might change protein effectiveness; c) High: variants assumed to have high (disruptive) impact on protein, probably causing protein truncation, loss of function or triggering nonsense mediated decay and including stop gained codons, splice donor variant and splice acceptor as well as start codon lost[36]. We also excluded intergenic (-no-intergenic) and intron (-no-intron) variants. For each variant category, we recorded the number of homozygous and heterozygous variants and summed the total number of variants. We estimated the total load and corrected for potential mapping biases arising from different sample types (i.e., batch effects associated with different datasets and unequal distance to outgroup) by calculating the ratio of deleterious variants (High and Moderate impact) to Synonymous SNPs, as previously described in Xue *et al.*[37]. We also calculated the individual realised load (i.e., total number of homozygous variants of category $i$ divided by twice the total number of segregating sites for category $i$) as described in

Mathur & DeWoody[65]. We compared the differences in load among populations using Wilcoxon signed-rank tests in R.

To take into account the frequency of variants in each population, we also calculated the $R_{xy}$ ratio for High and Moderate impact variants comparing the Crater (n = 10) vs Serengeti/Selous (n = 5) populations following Xue *et al.*[37]. We used a random number intergenic SNPs corresponding to the number of each impact type for standardisation, which makes $R_{xy}$ robust against sampling effects and population substructure[37]. We then estimated allele frequencies for intergenic, High and Moderate impact variants using PLINK and used custom scripts to calculate the $R_{xy}$. An $R_{xy}$ equal to 1 corresponds to no change in frequency between two populations, whereas $R_{xy}$ < 1 or >1 corresponds to a deficit/decrease or an excess/increase in frequency in population x (i.e., Crater) relative to population y (i.e., Serengeti/Selous), respectively. We used a jack-knife procedure in R to estimate the variance in the $R_{xy}$ ratio.

## Gene ontology

To identify putative genes associated with lower fitness in Crater lions and carrying High and Moderate impact variants, we used the Mouse Genome Informatics database (www.informatics.jax.org[39]) as well as literature searches to manually retrieve gene ontologies and mammalian phenotype information for each candidate gene.

## Population genomic simulations

The recent demographic history of the Crater lion population is well documented. However, it is crucial to assess whether our empirical results are consistent with the recent history of the population and assess the plausibility of our results interpretations. We thus performed forward genomic simulations in SLiM 4.0[41,42] to assess the potential effects of variable gene flow on the genomic variation of the Crater population from the 1950s until the present day and to predict the effects of future effective migration on genome-wide variation over the following 100 years.

We used a non-Wright-Fisher model (nonWF) which allows for overlapping generations and where each cycle in the simulation corresponds to a year. Also, the probability of an individual surviving from one year to the next is given by its absolute fitness, which ranges from 0 to 1 and which is determined by its genetic composition. Population size N is an emergent parameter controlled by carrying capacity (K) and is the outcome of a stochastic process of reproduction and viability selection. If N > K, the absolute fitness is rescaled downward by the ratio of K/N. Therefore, these models did not allow for population growth beyond K but instead allow for N to fluctuate around K.

To avoid the fitness of all individuals increasing to 1 in case of severe decline and to allow for viability selection and impacts of inbreeding depression, density-dependent selection was rescaled following Robinson *et al.*[66] by drawing the new individual fitness as min(K/N, 1.0).

We created scenarios recapitulating the population history of the Greater Serengeti Ecosystem (GSE) and Crater lion based on the demographic reconstructions from the PSMC and the recent population history and monitoring data[12,13]. To convert $N_e$ into N (i.e., K), we used a conservative $N_e/N_C = 0.1$[67]. After 400,000 years of burnin for a large ancestral population ($K_{Ancestral} = 50,000$), we modelled a $K_{GSE-Ancestral} = 5000$ between 5000 and 600 years Before Present (y. BP) and then modelled a decline a $K_{Historical-Present-GSE} = 2000$, between 600 y. BP and the present time[12]. We then modelled a population split between the GSE population and the Crater population 200 y. BP (i.e., based on the SMC + + divergence estimate; Supplementary Fig. 1) followed by a decline to $K_{1950s} = 100$ in ~1950 (Supplementary Data 8). We then modelled recent population fluctuations based on the well-documented demographic history of the Crater lion until the 2020s (Supplementary Data 3; Fig. 2c). From a population of ~100 in the 1950s, the population declined to nine adult females and one male due to an epizootic in 1962[12]. The following year, the population numbered ~15 lions which are the founders of the current population. We also modelled the arrival of breeding males: seven in 1964 and 1965, one in 1993, a coalition of four males in 2013, one male in 2015 and a coalition of

two males in 2018. For each time interval, we set K based on the census data (Supplementary Data 3, 8; Fig. 1c). We then ran the simulations for an additional 100 years period until the year 2120 to assess the future effects of gene flow by simulating 0, 1, 5 or 10 effective migrants (M) per decade and for K values of 50, 100, 150 or 200.

Reproduction was modelled with the simplifying assumptions of a first age at reproduction of 4 for males and females and last year of reproduction at 10 and 14 years old for males and females, respectively. We assumed a harem-like (polygamous) reproduction system with 69% and 78% of males allowed to reproduce for the GSE and Crater population, respectively[13]. Breeding occurred every two years[13]. We assumed a maximum of two litters (i.e., mating with up to two different females) per male per breeding cycle allowing for 40% and 60% of males mating with one and two females respectively to reflect the higher reproductive success of some males in a coalition (Supplementary Data 8). Assuming a litter size of 2.4 cubs (min=1; max=3)[13], we used offspring number probability values (i.e., weights) for each mating event of 0.2, 0.2 and 0.6 for 1, 2 and 3 cubs respectively. The cub sex ratio was set at 0.5 The model assumed a maximum longevity of 16 and 14 years for females and males, respectively and different age-specific mortality for each sex (Supplementary Data 8). We assumed an adult sex-ratio (4-14+ years old) of 2.75 (SD = 0.83) for the Crater population and of 2.65 (SD = 0.42) for the GSE population. When simulating gene flow, we assumed that females were philopatric[13] and only allowed males to immigrate and mate with local individuals during the year of arrival. In subsequent years, these males were joined to the pool of mating individuals with the same probability of mating as resident males.

We simulated 10 chromosomes containing each 500 genes of 1750 bp long following Robinson *et al.*[66]. We randomly generated deleterious (nonsynonymous) mutations in exonic regions at a ratio of 2.31:1 to neutral (synonymous) mutations[68]. For selection coefficients ($s$) of nonsynonymous mutations we used distributions based on estimates in humans[69] to model very strongly, strongly, moderately and weakly deleterious mutations as well as lethal mutations and using a gamma distribution a mean $s = -0.01314833$ and shape = 0.186 (Supplementary Data 8). For dominance coefficients ($h$), we assumed an inverse relationship between $h$ and $s$[70,71] with $h = 0.0$ for very strongly deleterious mutations ($s < -0.1$), $h = 0.01$ for strongly deleterious mutations ($-0.1 \leq s < -0.01$), $h = 0.1$ for moderately deleterious mutations ($-0.01 \leq s < -0.001$), and $h = 0.4$ for weakly deleterious mutations ($s > -0.001$). For neutral and lethal mutations $s$ was set to a fixed value of 0 and -1, respectively. We used a mutation rate of $1e^{-9}$ mutations/site/year (i.e., $4.5e^{-9}$ mutations/site/generation[31]) and assumed a generation time of five years[31]. For recombination rate, we assumed no recombination within genes, a rate of $1e^{-3}$ between genes, and free recombination between chromosomes.

Since we only simulated 5000 genes, we rescaled estimates of realised and masked load by a factor of 4 so that our estimates would correspond to a complete lion genome with ~20,000 protein coding genes (GCF_018350215.1). For the realised load, we used the exponential of the simulated value to the power of 4, since genetic load is multiplied across sites. For the masked load, we multiplied the simulated values by 4, since inbreeding load is summed across sites.

Summary statistics included population size (N), mean heterozygosity, mean $F_{ROH}$ ( >0.1 Mb), the mean number of each nonsynonymous mutation category (i.e., weakly to very highly deleterious deleterious) as well as mean realised load (i.e., reduction in fitness due to segregating and fixed deleterious mutations calculated multiplicatively across sites; 1 - estimated Fitness) and mean masked load (i.e., the quantity of recessive deleterious variation concealed in heterozygotes, measured as the sum of diploid lethal equivalents) as estimated in Kyriazis *et al.*[7]. We estimated all statistics based on a sample of 15 individuals every 2 years. For all K-M combinations we ran a total of 80 replicates each starting with a different seed. We then plotted indices of diversity through time, changes in the number of deleterious alleles and the 5% change in genetic indices over 100 years in R.

## Reporting summary

Further information on research design is available in the Nature Portfolio Reporting Summary linked to this article.

## Data availability

Resequencing data (ENA BioProject): PRJEB80542 (this study), PRJNA611920, PRJNA182708, PRJNA16726, PRJNA854353, PRJNA684344. The provided accession codes are also available at NCBI. See Supplementary Data 4 for estimates of Heterozygosity, inbreeding and genetic load.

## Code availability

Code for data processing and analysis, and simulation are deposited to Github: https://github.com/ndussex/Crater_lion_genomics.

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

## Acknowledgements
We thank the Government of Tanzania, TAWIRI, and the Ngorongoro Conservation Area Authority for their support, and especially acknowledge their veterinarian, late Dr. A.R. Nyaki. We also acknowledge sequencing support from the Swedish National Genomics Infrastructure (NGI) at the Science for Life Laboratory in Uppsala and Stockholm supported by the Swedish Research Council and the Knut and Alice Wallenberg Foundation. The computations and data handling was enabled by resources provided by the National Academic Infrastructure for Supercomputing in Sweden (NAISS) and the Swedish National Infrastructure for Computing (SNIC) partially funded by the Swedish Research Council through grant agreements no. 2022-06725 and no. 2018-05973. The sample collection was supported by the Swedish Research Council through grant no. 2014-03382, and by various funds raised by KopeLion Inc. We thank the two anonymous reviewers for the constructive comments and suggestions.

## Author contributions
G.S. and N.D. conceived and designed the study. I.J. collected the NCA samples. G.S. acquired and generated genomic data. N.D. analysed the data. G.S., N.D., I.J., T.v.d.V., B.M.K., E.E.M., A.N. and C.P. interpreted the data. G.S. provided funding. N.D. wrote the manuscript with input from the other authors.

## Funding

## Competing interests
The authors declare no competing interests.
