## [Transparent Peer Review file · Communications Biology]

Constraints to gene flow increase the risk of genome erosion in the Ngorongoro Crater lion population

Corresponding Author: Dr Nicolas Dussex

This manuscript has been previously reviewed at another journal. This document only contains information relating to versions considered at Communications Biology.

Version 0:

Reviewer comments:

Reviewer #1

(Remarks to the Author)

Thank you for the opportunity to review the manuscript COMMSBIO-24-7958-T entitled “Strong male territoriality increases the risk of genome erosion in Ngorongoro Crater lions.” I enjoyed reading the paper and find the topic particularly relevant today, in our ever-changing world where wildlife populations are forced into small pockets of (potentially isolated) suitable habitat. In this particular study, the authors sequenced 15 lions from the Tanzanian Ngorongoro Crater and surrounding Serengeti to look at the effect of two subsequent population bottlenecks. The manuscript is well written and structured, so from that perspective I have little to no comments. However, the data has often been overinterpreted and there are quite a few misconceptions and inconsistencies in the manuscript. Most importantly, the authors focus their results on recent and past population declines (i.e. bottlenecks) but show no actual evidence that immigration into the Crater is reduced, nor provide any data on male territoriality, which they suggest is stronger in the Crater, thus worsening the population isolation. Therefore, the title is very misleading. The author also talk about gradual changes in genetic diversity and purging, which is impossible when you do not have generational data. The comparison of your data to other areas (Botswana and South Africa) is also misleading, consider the distance and sample size difference. My concern is that the authors have tried to oversell their results, which has been the root cause of such errors. Luckily, fifteen whole genomes is interesting enough to allow for a scientifically solid paper, but then some vigorous changes need to be applied before considering this manuscript for publication. Therefore, I recommend major revisions, which I elaborate in a point-by-point manner below.

Line 46: You state: “genetic load may not substantially change individual fitness if there is quick recovery” making it sounds like some epigenetic mechanism that can change during the course of an individual’s lifetime. Consider rephrasing the sentence for clarity.

Line 48: It might be useful to explain to the reader what purging is.

Line 58: I find the “(e.g. kakapo) quite odd here – used differently from the other brackets in the same sentence, which serve to explain a term. Not everyone may be familiar with the kakapo bird or its life history.

Line 62-64: This sentence seems entirely unrelated: critically endangered species with low reproductive rates (lions are neither)

Line 66: “due to sufficient prey abundances”; so does this sentence imply that there is suboptimal habitat and lots of prey just outside the Crater but not inside? And how do you propose that this will isolate this lions in the Crater?

Line 68: “only little immigration”. Considering that this is the central point of your study, it would be good to know what is meant by little. For instance, only six lions over the past 10 years (2 generations) would still be ten percent of the entire population, which would be significant.

Line 68: Remove “Indeed”

Line 70: I was so surprised by the statement that there is higher competition in the Crater than elsewhere (which supposedly causes reproductive isolation in the Crater) that I went to check the reference. It turns out that this source can’t be verified because it’s in preparation. Hence, this data won’t come out before this publication. Considering the pretty extreme nature of the statement, I recommend finding other references or removing the sentence entirely. Or at least explain very well why mating opportunities are different / more limited in the Crater, when compared to other lions, and why this would isolate the

population. It would also be good to explain how inbreeding avoidance, which is so common in mammals (particularly social breeders), comes into play.

Line 81: Why did it decline again the 90s?

Line 83: Rephrase sentence

Line 87: Why include genomes from Botswana and South Africa?

Line 89: Here you state "recent population decline" but the results (line 99) starts with "past population history". Beside this conflict, I don't see how these relate to the title.

Line 91: You state that your results show that reduced immigration into the Crater induced genomic erosion, but your results show absolutely no proof of that. All it does is show the effect of a population bottleneck (and founder effect)

Line 93: Biologists usually do these calculations per generation, instead of per decade, but okay; however you prefer it in this case.

Line 100: You start the results with structure analyses, but haven't clarified why you consider this necessary to answer your research question. More worryingly: you base isolation on mtDNA markers, which is completely unsuitable to look at recent population structure because it is not part of introgression and recombination (nor representative of male dispersal). Furthermore, most mitochondrial genotypes are likely to have gone lost during the bottleneck, which means that any observed difference with the surrounding area is due to drift (and not isolation).

Line 103: Structure analysis is often overinterpreted and does not properly deal with introgression. For instance, you won't ever see a case where $K=1$ is the result, so again, additional analyses are needed to conclude that Crater lions are isolated.

Line 105: This sentence is too important to write off with "however"

Line 110: As for Figure 1: the sample names "Lion01, Lion03" etc are uninformative. It'd be better to name them according to location. I can't even tell from the structure analyses which ones are Crater lions. In 1a: what are Ndulu and Endulen? And in 1c: why is there a cheetah in there? And where did South Africa go? 1d. I also can't tell from this PCA which ones are Crater lions – maybe give Crater ones a special shape?

Line 116: Again, I'm not entirely sure why you think PSMC is necessary to answer your research questions. There are three paragraphs about past climatic events, which is interesting but seemingly off-topic.

Line 132: You conclude genetic isolation again here, but isn't a bottleneck effect more likely? Figure 2b surely points that way

Line 148: Before you start to make these comparisons, it is important to remove any related individuals from the Crater (if you haven't already). You mention in the methods that you selected presumed-to-be unrelated individuals (based on field observations) but this can and should actually be tested with the genomic data.

Line 152: I agree with this statement: long ROHs and population bottlenecks are the perfect match. But once again: please reconsider the title.

Line 157: "suggesting that the parameters provide robust estimates" – why wouldn't they?

Line 161: Figure 3: You say 20 lion genomes, but in the figure I count 15.

South Africa is just 1 genome, and knowing South Africa it's probably from a private reserve that stems from a reintroduction, which is likely to be mildly inbred as well. This comparison is thus pretty much pointless.

What's more: the 5X sequencing has only been mentioned now and is pretty significant / concerningly low. Sequencing depth has a huge impact on measures of genetic diversity and structure, due which an ANGSD pipeline is normally recommended (based on genotype likelihoods). Otherwise just stick to the genomes that have 14X depth.

Line 181: South Africa is not outbred

Line 185: Didn't you say in the introduction that purging occurs only where there is a slow reduction in population size? The lion's bottleneck was pretty rapid – so perhaps dispersal into the Crater may actually be a more viable alternative hypothesis.

Line 186: "other populations"; please clarify which ones you mean here. I think a comparison to South Africa and Botswana are non-informative due to sample size, among other factors.

Line 190: "As theory, empirical data ..."; if there is empirical data, it's no longer a theory

You say that the early stage of a decline is characterized by such and such. But when did your sampling take place? I presume pretty recently, so not in those 'early stages', i.e. 1962

Line 195: "The patterns observed" followed by the comparison to gorilla is pretty invalid, because you actually don't have generational data: you just have one snapshot of 10 genomes. You can't say anything about a reduction of deleterious alleles through time, because you don't know what the situation was like before or after 1962.

Line 201: Like the previous comment: you definitely can't say anything about purging (which requires generational data), let alone when you look at one lion in a private park in South Africa – or "likely" a zoo?). Also, which genome grouped with SA lions? Based on mtDNA? I don't recall reading this.

Line 237: So there was actually effective immigration into the Crater, twice, and you mention this only now. This seems rather important and in stark contrast to your persistent conclusion that Crater lions are isolated.

Line 246: Again: very important and in stark contrast to the title.

Line 250: What are Very Strongly deleterious variants? How can deleterious alleles not differ after a bottleneck, but Very Strong ones be a direct effect thereof?

Line 252: But purging and bottlenecks don't seem to go together, I've been told

Line 301: This statement really needs to be taken out before this paper can be considered for publication. There is no scientific base for this whatsoever. In fact, it conflicts entirely with your findings, which seems to show low genetic load, likely due to effective dispersal that occurred in 1964-65 and 2013-18. These simulations are meaningless here; you can essentially just feed the programme whatever you like.

Like 304: "Taken together, our results challenge the notion": I disagree that your results do this at all. In the end, you have 10 genomes from the Crater, from one moment in time. Which is just fine (it can give plenty of interesting results); but it's crucial not to oversell them.

Line 320/27: Here, you pretty much confirm the point I wanted to make: it's human-lion conflict and habitat fragmentation that are the problem: NOT male territoriality.

Line 341: Here you should mention the actual genetic relatedness of your samples.

Line 344: Here you should mention the targeted and realized sequencing depth of your genomes

Line 372: You mention "gene flow" in the title, but only give structure-focused analyses. Structure could result from lack of gene flow, surely, but also from differences in genomic diversity. Maybe it's worth adding EEMS, introgression and recombination rates, and genome-wide estimates of the frequency of private alleles

Line 386: When did SA go, and why include Botswana? Including outliers like this can actually cause a polarizing effect on the structure analyses

Line 470: What are variants of category i?

Line 494: What is meant by genome-informed? Do you upload your WGS data?

Reviewer #2

(Remarks to the Author)

This manuscript described a study of inbreeding, genetic load, and population structure of a population of East-African lions focused on a semi-isolated population in Ngorongoro Crater. Empirical analysis of 20 lion resequenced genomes was paired with forward-in-time simulations of the population in recent history to explore current and likely future consequences of a recent demographic bottlenecks. Principle findings were that current inbreeding was higher (estimated at 2x higher) than that back-projected to the time of the 1962 epizootic and that genetic load, particularly of the most deleterious alleles, was higher than observed in a neighboring population. For the most part, the manuscript seemed sound and was clearly written. I think the core of the results are relevant to the conservation of an iconic species and provide a novel and interesting insights into a lion metapopulation impacted by anthropogenic changes of the past two centuries.

However, I found the title and statements related to the title in the Abstract and Conclusion to be misleading; I think the title should be changed to more appropriately describe what the study actually investigated. As best I could tell, this study did not investigate effects of strong male territoriality (or any other life-history trait) on genomic erosion. Rather, it simply investigated genomic erosion in a species that happens to have such a life-history trait. There was no comparison or specific investigation of how reproductive skew, etc., affected inbreeding or the accumulation, fixation, and purging of deleterious alleles. I agree with the authors' argument for their expectation (i.e., assumption) that the particular life history of lions should render it especially vulnerable to genetic erosion, but that is not the same thing as having investigated it.

I also have included several minor comments below and on a markup version of the manuscript, including several editorial suggestions on the latter.

Lines 20-21: I think this statement should be deleted as it misleadingly suggests the study will investigate effects of a life-history trait on genetic erosion.

Lines 58-61: please explain this more clearly. It was not at all apparent to me why generation time and litter size per se should affect genetic drift, which depends primarily (only?) of genetic effective population size and is typically measured in terms of rate of fixation (or change in allele frequency) per generation.

Line 104: I disagree with the interpretation that $K = 2$ was best supported regardless of the CVE, which is simply a guide (and should be included on the figure that was referenced in association with that statement). The bar graph at $k = 3$ resulted in the cleanest population structure (i.e., nearly 0 admixture), whereas $K = 2$ showed substantial admixture in 5 of the lions. Deleting that statement would not affect your other results, and would be more consistent with your inclusion of several values of K on the figure, which I agree are informative.

Lines 116-122 and Figure 2a: I think the PSMC, this text, and figure should be removed. They are irrelevant to the rest of the paper and apparently duplicate previous results in any case.

Line 194: The statement that neutral mutations tend to drift to fixation seems incorrect as they are no more likely to drift to fixation (i.e., frequency of 100%) than they are to extinction (frequency of 0%).

Lines 197-198: I think you have made a fallacious inference here. Your observation was only that Crater lions had more "High impact variants" than did another population, but that observation does not in and of itself imply anything about which population experienced a change in that quantity or in which direction. It seems at least as likely (possibly more so) that the other population lost more of its High impact variants (to purging, drift, or both), which would cause the same observed difference.

Line 200: you state that the South African individual had the lowest load, but that is not supported by figure 2b.

Lines 299-301: I disagree with this statement. You simply reported stats on this population, but did not directly relate them to the life-history or behavior of lions per se.

Lines 303-305: This does not make sense as a general conclusion. "Demographic recovery" has a system-specific meaning. If your conclusion is that the upward trajectory of the Crater lion population is not sufficient to safeguard it against inbreeding depression, fine, but that also then implies that it has not recovered sufficiently (demographically or otherwise).

Lines 324-328: I would delete this. It is a throwaway paragraph and not uniquely about this study.

Line 344: please provide more information on the PCR-free library preparation protocol.

Reviewer comments:

Reviewer #1

(Remarks to the Author)

I appreciate the changes and believe the manuscript has improved substantially. I consider the manuscript to be currently suitable for publication. However, there are still some issues that need attention, which I elaborate in the attachment. Because these suggestions are very minor, I do not require another reviewer's report.

Reviewer #2

(Remarks to the Author)

I think the authors addressed all of my most significant issues and the present draft is much improved. I have only minor editorial comments on this draft, which, if the authors accept, will make their paper clearer, most importantly in the Introduction.

Reviewers' comments:

Reviewer #1 (Remarks to the Author):

Thank you for the opportunity to review the manuscript COMMSBIO-24-7958-T entitled “Strong male territoriality increases the risk of genome erosion in Ngorongoro Crater lions.” I enjoyed reading the paper and find the topic particularly relevant today, in our ever-changing world where wildlife populations are forced into small pockets of (potentially isolated) suitable habitat. In this particular study, the authors sequenced 15 lions from the Tanzanian Ngorongoro Crater and surrounding Serengeti to look at the effect of two subsequent population bottlenecks. The manuscript is well written and structured, so from that perspective I have little to no comments. However, the **data has often been overinterpreted and there are quite a few misconceptions and inconsistencies in the manuscript**. Most importantly, the authors focus their results on recent and past population declines (i.e. bottlenecks) but show no actual evidence that immigration into the Crater is reduced, nor provide any data on male territoriality, which they suggest is stronger in the Crater, thus worsening the population isolation. Therefore, the **title is very misleading**. The author also talk about gradual changes in genetic diversity and purging, **which is impossible when you do not have generational data**. The comparison of your data to other areas (Botswana and South Africa) is also misleading, consider the distance and sample size difference. My concern is that the authors have tried to oversell their results, which has been the root cause of such errors. Luckily, fifteen whole genomes is interesting enough to allow for a scientifically solid paper, but then some vigorous changes need to be applied before considering this manuscript for publication. Therefore, I recommend major revisions, which I elaborate in a point-by-point manner below.

>>> We thank the reviewer for their constructive comments, which have allowed us to greatly improve the manuscript. we have toned down our interpretation of the results and reframed our conclusion around the impact of bottleneck and habitat-fragmentation on genome erosion in Crater lions. Nevertheless, we have now added more details on the evidence supporting higher male territoriality in the Crater in the introduction (Packer 2023, Chapter 10). While only speculative, we believe it is worth mentioning the potential role of male territoriality in the pattern of genome erosion observed.

However, we respectfully disagree that only generational (i.e. temporally-spaced) data allows to show evidence of purging. While temporal data is ideal to test for purging ‘in real time’ as we have done previously (see Dussex et al. 2021, 2023; von Seth et al. 2021, 2022), a number of studies have used comparative analyses with modern data alone to test for evidence of purging (e.g. Ibex, Grossen et al. 2020; rattlesnake, Mathur et al. 2023; Montezuma quail, Mathur & deWoody 2021; Svalbard reindeer, Dussex et al. 2023; Moose, Kyriazis et al. 2023; Orkney voles, Wang et al. 2023). Furthermore, simulations are a valuable alternative and addition to temporally-spaced data as they allow to validate the interpretation (in this case, lack

of purging in Crater lions) of empirical data (see Kyriazis et al. 2023, American Naturalist, for a review). Importantly, the parameters used in these models are not approximations but have for most of them been measured in the field and over several decades by co-authors of this manuscript who are lion specialists (Ingela Jansson, Craig Packer).

Finally, we agree that the sample size and geographical representation is patchy and can limit the scope of the study. However, as shown in similar studies (e.g. de Manuel et al. 2020), this can provide informative points of comparison and clues on past demography for instance (e.g. using Runs of Homozygosity). We thus feel that they should not be removed from the manuscript altogether. Furthermore, most of the results and discussion focuses on the populations with the largest sample sizes (i.e., Crater and Greater Serengeti).

Line 46: You state: “genetic load may not substantially change individual fitness if there is quick recovery” making it sounds like some epigenetic mechanism that can change during the course of an individual’s lifetime. Consider rephrasing the sentence for clarity.

>>> We have rephrased this sentence on l. 49-50 which now reads : *‘Similarly, the amount of genetic load may not substantially change and thus impact individual fitness in future generations if there is a quick population recovery.’*.

Line 48: It might be useful to explain to the reader what purging is.

>>> We have added a brief explanation of purging on l. 50-53 and state that *‘In contrast, long-term declines characterised by gradual increases in inbreeding may induce a reduction in deleterious variation through purifying selection, in a process referred to as purging, whereas rapid declines may induce an increase in the frequency of deleterious variants’*.

Line 58: I find the “(e.g. kakapo) quite odd here – used differently from the other brackets in the same sentence, which serve to explain a term. Not everyone may be familiar with the kakapo bird or its life history.

>>> We have clarified this statement on l. 64.

Line 62-64: This sentence seems entirely unrelated: critically endangered species with low reproductive rates (lions are neither)

>>> We agree with the reviewer and have removed this sentence.

Line 66: “due to sufficient prey abundances”; so does this sentence imply that there is suboptimal habitat and lots of prey just outside the Crater but not inside? And how do you propose that this will isolate this lions in the Crater?

>>> Yes, this is what we meant. This sentence suggests that the limited opportunities for pride occupancy due to suboptimal habitat and resources outside the Crater may be contribute the population's isolation. We have now clarified and rephrased this sentence on l. 67-71: '*...seems to function as an ecologically isolated population. The Crater offers high prey abundance and minimal human threats whereas the surrounding landscape is a multi-use area shared with pastoralists and their livestock, where natural prey densities are lower and more variable, and where human-lion conflicts are frequent. Thus, opportunities for pride establishment outside the Crater are limited*¹²⁻¹⁴.'

Line 68: "only little immigration". Considering that this is the central point of your study, it would be good to know what is meant by little. For instance, only six lions over the past 10 years (2 generations) would still be ten percent of the entire population, which would be significant.

>>> This is a fair point. We now clarify this statement on l. 72-73.

Line 68: Remove "Indeed"

>>> We have removed 'indeed'.

Line 70: I was so surprised by the statement that there is higher competition in the Crater than elsewhere (which supposedly causes reproductive isolation in the Crater) that I went to check the reference. It turns out that this source can't be verified because it's in preparation. Hence, this data won't come out before this publication. Considering the pretty extreme nature of the statement, I recommend finding other references or removing the sentence entirely. Or at least explain very well why mating opportunities are different / more limited in the Crater, when compared to other lions, and why this would isolate the population. It would also be good to explain how inbreeding avoidance, which is so common in mammals (particularly social breeders), comes into play.

>>> We have now removed the reference to the 'in prep.' manuscript and have expanded on the territoriality argument based on Chapter 10 of 'Packer, C. The Lion: Behavior, Ecology, and Conservation of an Iconic Species. (Princeton University Press, 2023).' We now provide more details supporting the argument of territoriality in male Crater lions and now only suggest that it could also contribute to the genetic isolation of the population.

We have also added a sentence on inbreeding avoidance on l. 81-83.

Line 81: Why did it decline again the 90s?

>>> We have now added some information on l. 94-96. The sentence now reads: '*After a period of recovery, the population declined again in the late 90s although the real cause is unknown*²³ and in 2001, from a combination of tick-borne disease and canine distemper virus (CDV)²⁴, with the number of lions close to 50 individuals over the past ~30 years'.

Line 83: Rephrase sentence

>>> We have rephrased and split this sentence in two sentences on l. 99-100.

Line 87: Why include genomes from Botswana and South Africa?

>>> We used those genomes, as well as two other published genomes from Tanzania as a point of comparison and to increase the scope of our population structure analyses. We have now clarified this sentence on l. 104-110.

Line 89: Here you state “recent population decline” but the results (line 99) starts with “past population history”. Beside this conflict, I don’t see how these relate to the title.

>>> We have now rephrased the title. We have also removed ‘past’ on l. 122 to remove this confusion.

Line 91: You state that your results show that reduced immigration into the Crater induced genomic erosion, but your results show absolutely no proof of that. All it does is show the effect of a population bottleneck (and founder effect)

>>> We have rephrased this sentence on l. 111-113 and now state that ‘*Genomic data suggest that the recent decline induced an increase in genome erosion and realised load in the Crater population while simulations suggest that reduced male immigration could have potentially contributed to the observed pattern.*’

Line 93: Biologists usually do these calculations per generation, instead of per decade, but okay; however you prefer it in this case.

>>> We take note of this comment.

Line 100: You start the results with structure analyses, but haven’t clarified why you consider this necessary to answer your research question. More worryingly: you base isolation on mtDNA markers, which is completely unsuitable to look at recent population structure because it is not part of introgression and recombination (nor representative of male dispersal). Furthermore, most mitochondrial genotypes are likely to have gone lost during the bottleneck, which means that any observed difference with the surrounding area is due to drift (and not isolation).

>>> We have now added a justification for performing population structure analyses on l. 125-127.

>>> The reviewer raises a fair point regarding the mitochondrial phylogeny and we have now removed the mitochondrial phylogeny. However, we do not base the isolation solely on this analysis and provide an admixture plot and PCA and also added an

estimation of the divergence time between Ngorongoro Crater and Serengeti plains genomes.

Line 103: Structure analysis is often overinterpreted and does not properly deal with introgression. For instance, you won't ever see a case where $K=1$ is the result, so again, additional analyses are needed to conclude that Crater lions are isolated.

>>> We have clarified the presentation of the admixture results and have provided the Cross Validation values, as requested by R2's comments on l. 128-131.

Line 105: This sentence is too important to write off with "however"

>>> We have removed 'however' on l. 131.

Line 110: As for Figure 1: the sample names "Lion01, Lion03" etc are uninformative. It'd be better to name them according to location. I can't even tell from the structure analyses which ones are Crater lions. In 1a: what are Ndulu and Endulen? And in 1c: why is there a cheetah in there? And where did South Africa go? 1d. I also can't tell from this PCA which ones are Crater lions – maybe give Crater ones a special shape?

>>> We have relabelled lion genomes in the admixture plot (now Fig. 1c) and removed the mitochondrial tree. In the legend of the PCA (now Fig. 1b), we have clarified the legend, indicating that Endulen, Ndutu and Tanzania are grouped into Greater Serengeti. We have also removed the admixture graph. We also note that Cheetah was there as an outgroup.

Line 116: Again, I'm not entirely sure why you think PSMC is necessary to answer your research questions. There are three paragraphs about past climatic events, which is interesting but seemingly off-topic.

>>> We understand the reviewer's point. However, we feel it is important to keep the PSMC to put the population history of lions into a broader temporal context. Furthermore, the previous lion genomics paper from de Manuel et al. (2020), did not perform a PSMC reconstruction for Crater lions.

Line 132: You conclude genetic isolation again here, but isn't a bottleneck effect more likely? Figure 2b surely points that way

>>> What we mean is that the main drivers of this isolation is the bottleneck and that habitat fragmentation outside the Crater may exacerbate this genetic isolation. We have rephrased this sentence on l. 134-137. such as: *'The timing of this divergence is thus consistent with the recent bottleneck and increase in habitat fragmentation over the past 200 y. BP which both contribute to the genetic isolation of the population.'*

Line 148: Before you start to make these comparisons, it is important to remove any related individuals from the Crater (if you haven't already). You mention in the methods that you selected presumed-to-be unrelated individuals (based on field observations) but this can and should actually be tested with the genomic data.

>>> We have estimated relatedness and mention it in the methods on l. 365-367.

Line 152: I agree with this statement: long ROHs and population bottlenecks are the perfect match. But once again: please reconsider the title.

>>> We thank the reviewer for this comment.

Line 157: "suggesting that the parameters provide robust estimates" – why wouldn't they?

>>> We have removed this statement.

Line 161: Figure 3: You say 20 lion genomes, but in the figure I count 15.

>>> We have corrected this and refer to 16 lions in Figure 3.

South Africa is just 1 genome, and knowing South Africa it's probably from a private reserve that stems from a reintroduction, which is likely to be mildly inbred as well. This comparison is thus pretty much pointless.

>>> We respectfully disagree. Even though a sample size of one is not statistically powerful, even a single genome can provide relevant information. For instance, ROH distribution can inform about the past population history of a population by allowing to estimate the timing of inbreeding events. Thus, we feel that having this point of comparison is still informative.

What's more: the 5X sequencing has only been mentioned now and is pretty significant / concerningly low. Sequencing depth has a huge impact on measures of genetic diversity and structure, due which an ANGSD pipeline is normally recommended (based on genotype likelihoods). Otherwise just stick to the genomes that have 14X depth.

>>> This is an error on our part. We were referring to 16 high coverage genomes and we did not downsample to 4X but to 14X. This has been amended on l. 187, 396 and 436.

Line 181: South Africa is not outbred

>>> The reviewer is correct. We have now added a reference to the Serengeti and Selous populations on l. 206.

Line 185: Didn't you say in the introduction that purging occurs only where there is a slow reduction in population size? The lion's bottleneck was pretty rapid – so perhaps dispersal into the Crater may actually be a more viable alternative hypothesis.

>>> In the introduction, we state that purging is facilitated when there is a gradual decline. However, some purging will occur anyways, even in the early stages of a decline and assuming that there is some increase in inbreeding, although a longer period of decline and longer exposure of deleterious alleles in homozygous state will facilitate this. In fact, purging (i.e. purifying selection) happens all the time, but the speed and efficiency of this process will depend on population size (i.e. N_e) and selection coefficient of mutations, as we discussed in Dussex et al. (2023, TREE). Furthermore, effective dispersal will not cause a purging effect, but instead the opposite through the introduction of new deleterious alleles (e.g. Isle Royale wolf, Robinson et al. 2019, Science; Dussex et al. 2023, TREE).

Line 186: “other populations”; please clarify which ones you mean here. I think a comparison to South Africa and Botswana are non-informative due to sample size, among other factors.

>>> Botswana is not included in this analysis and we only compare Crater to Serengeti, Selous and South Africa. We have clarified this statement on l. 211.

Line 190: “As theory, empirical data ...” ; if there is empirical data, it's no longer a theory. You say that the early stage of a decline is characterized by such and such. But when did your sampling take place? I presume pretty recently, so not in those ‘early stages’, i.e. 1962

>>> We make a general statement and refer here to three elements: theory, empirical data and simulations. This statement is based on a number of empirical studies and key reviews on the topic (e.g. Hedrick & Garcia-Dorado 2016; Dussex et al. 2023). The sampling date of our samples is not relevant to this statement.

Line 195: “The patterns observed” followed by the comparison to gorilla is pretty invalid, because you actually don't have generational data: you just have one snapshot of 10 genomes. You can't say anything about a reduction of deleterious alleles through time, because you don't know what the situation was like before or after 1962.

>>> We respectfully disagree with this comment. While temporal data is ideal to test for purging as we have done in the past (see Dussex et al. 2021, von Seth et al. 2021, 2022), a growing number of studies have used comparative analyses with modern data alone to test for evidence of purging (e.g. Ibex, Grossen et al. 2020; rattlesnake, Mathur et al. 2023; Montezuma quail, Mathur & deWoody 2021).

Line 201: Like the previous comment: you definitely can't say anything about purging (which requires generational data), let alone when you look at one lion in a private park in South Africa

– or “likely” a zoo?). Also, which genome grouped with SA lions? Based on mtDNA? I don’t recall reading this.

>>> We respectfully disagree with this comment as well. We cite de Manuel et al. (2020, PNAS) to support the statement that this genome is most likely from South Africa. We have clarified this statement on l. 228-231.

Line 237: So there was actually effective immigration into the Crater, twice, and you mention this only now. This seems rather important and in stark contrast to your persistent conclusion that Crater lions are isolated.

>>> We now mention these immigration events in the introduction on l. 97-98.

Line 246: Again: very important and in stark contrast to the title.

>>> We now mention these immigration events in the introduction on l. 97-98.

Line 250: What are Very Strongly deleterious variants? How can deleterious alleles not differ after a bottleneck, but Very Strong ones be a direct effect thereof?

>>> The explanation is given in the following sentence in the text on l. 282. Variants with the highest selection coefficient and thus highest impact on fitness will be purged more quickly, early and efficiently during the bottleneck, whereas those with lower selection coefficient are more likely to get fixed by drift over the long term due to their lower impact on fitness (see Dussex et al. 2023, TREE for a review on the dynamics of genetic load).

Line 252: But purging and bottlenecks don’t seem to go together, I’ve been told

>>> We are not sure we understand what the reviewer is referring to here. In fact purging is an ongoing process whose effectiveness will vary in function of the effective population size, inbreeding, strength of bottleneck and the selection coefficients of deleterious mutations. It is true that during a sudden and severe bottleneck, drift will be strong and may tend to lead to an increase in frequency of deleterious alleles. However, the most deleterious mutations will be purged more quickly or in the earliest stage of the bottleneck as inbreeding increases. This point is illustrated in Fig. 3 of Dussex et al. (2023, TREE) We have clarified this statement on l. 282. and state that ‘*This is not entirely surprising since the most deleterious variants should be purged relatively early during a bottleneck through purifying selection as they have the most impact on fitness*’.

Line 301: This statement really needs to be taken out before this paper can be considered for publication. There is no scientific base for this whatsoever. In fact, it conflicts entirely with your findings, which seems to show low genetic load, likely due to effective dispersal that

occurred in 1964-65 and 2013-18. These simulations are meaningless here; you can essentially just feed the programme whatever you like.

>>> We have amended this statement on l. 326-327 in light of R1 and R2's comments. However, we respectfully disagree with the reviewer on this point. First, as stated above, we have provided more information of high male territoriality in the Crater, in the introduction on l. 67-83. Secondly, the Crater population has a higher total and realised load compared to the other populations (Fig. 4). Importantly, key studies and reviews have shown that effective dispersal into an inbred population actually increases genetic load and its expression (see Robinson et al. 2019, *Science*; Dussex et al. 2023, *TREE*). Even if a population is not inbred, gene flow from a distinct population is more likely to increase total load than reduce it. What is also crucial, is not only the amount of load (correlated to N_e) but the distribution of allele frequencies of deleterious variants. Third, we disagree that simulations are meaningless. In fact, there has been repeated calls to use simulations in population genomics studies (e.g. Haller & Messer, *American Naturalist*, 2023; Kyriazis et al. *American Naturalist*, 2023) as well as many empirical studies (e.g. Kyriazis et al. 2023, *MBE*; Dussex et al. 2021, 2023; Speak et al. 2024, *MolEcol*; Dussex et al. 2024, *SEPA* report) to assess the interpretation of results or predict future genomic trajectories of populations. Importantly, the parameters used for our simulations were informed by ecological field studies by Craig Packer, Ingela Jansson, and Göran Spong in the Crater, Serengeti and Selous.

Like 304: "Taken together, our results challenge the notion": I disagree that your results do this at all. In the end, you have 10 genomes from the Crater, from one moment in time. Which is just fine (it can give plenty of interesting results); but it's crucial not to oversell them.

>>> We have also rephrased this statement on l. 329-331.

Line 320/27: Here, you pretty much confirm the point I wanted to make: it's human-lion conflict and habitat fragmentation that are the problem: NOT male territoriality.

>>> We have toned down our interpretation throughout the manuscript.

Line 341: Here you should mention the actual genetic relatedness of your samples.

>>> We have estimated pairwise relatedness and mention it on l. 365-367.

Line 344: Here you should mention the targeted and realized sequencing depth of your genomes

>>> We have added this information on l. 372.

Line 372: You mention "gene flow" in the title, but only give structure-focused analyses. Structure could result from lack of gene flow, surely, but also from differences in genomic

diversity. Maybe it's worth adding EEMS, introgression and recombination rates, and genome-wide estimates of the frequency of private alleles

>>> We have now removed the mention of gene flow in this title since we are only referring to population structure and isolation.

Line 386: When did SA go, and why include Botswana? Including outliers like this can actually cause a polarizing effect on the structure analyses

>>> This was an omission on our part. However, as requested by the editor, we have now completely removed the Admixture graph analysis.

Line 470: What are variants of category i?

>>> This symbol refers to any category considered. We have now italicised the symbol for clarification.

Line 494: What is meant by genome-informed? Do you upload your WGS data?

>>> No, we have not used genetic data in the simulations. We have now replaced '*genome-informed*' with '*genomic*' for clarity.

Reviewer #2 (Remarks to the Author):

please see also attached pdf for comments embedded in manuscript file

This manuscript described a study of inbreeding, genetic load, and population structure of a population of East-African lions focused on a semi-isolated population in Ngorongoro Crater. Empirical analysis of 20 lion resequenced genomes was paired with forward-in-time simulations of the population in recent history to explore current and likely future consequences of a recent demographic bottlenecks. Principle findings were that current inbreeding was higher (estimated at 2x higher) than that back-projected to the time of the 1962 epizootic and that genetic load, particularly of the most deleterious alleles, was higher than observed in a neighboring population. For the most part, the manuscript seemed sound and was clearly written. I think the core of the results are relevant to the conservation of an iconic species and provide a novel and interesting insights into a lion metapopulation impacted by anthropogenic changes of the past two centuries.

However, I found the title and statements related to the title in the Abstract and Conclusion to be misleading; I think the title should be changed to more appropriately describe what the study actually investigated. As best I could tell, this study did not investigate effects of strong

male territoriality (or any other life-history trait) on genomic erosion. Rather, it simply investigated genomic erosion in a species that happens to have such a life-history trait. There was no comparison or specific investigation of how reproductive skew, etc., affected inbreeding or the accumulation, fixation, and purging of deleterious alleles. I agree with the authors' argument for their expectation (i.e., assumption) that the particular life history of lions should render it especially vulnerable to genetic erosion, but that is not the same thing as having investigated it.

>>> We thank the reviewer for their comments. We agree with them that the title can be misleading and it was only a speculation that male territoriality could have contributed to genome erosion in Crater lions. We have now added more information on the evidence supporting strong male territoriality in the Crater (Packer 2023) on l. 74-81. We also focus more on isolation and habitat fragmentation as a driver of the observed pattern and have toned down our interpretation throughout the text.

I also have included several minor comments below and on a markup version of the manuscript, including several editorial suggestions on the latter.

>>> We have taken notes of the comments on the PDF version of the manuscript and amended the manuscript accordingly.

Lines 20-21: I think this statement should be deleted as it misleadingly suggests the study will investigate effects of a life-history trait on genetic erosion.

>>> We have edited this statement and now state '*However, constrained gene flow can hamper genomic recovery.*'

Lines 58-61: please explain this more clearly. It was not at all apparent to me why generation time and litter size per se should affect genetic drift, which depends primarily (only?) of genetic effective population size and is typically measured in terms of rate of fixation (or change in allele frequency) per generation.

>>> It is important to point out that N_e reflects the strength of drift. So, stating that genetic drift depends on effective population size is a bit circular. Nevertheless, we have now removed these two sentences as they were misleading, since lions have for a mammal a rather large reproductive output.

Line 104: I disagree with the interpretation that $K = 2$ was best supported regardless of the CVE, which is simply a guide (and should be included on the figure that was referenced in association with that statement). The bar graph at $k = 3$ resulted in the cleanest population structure (i.e., nearly 0 admixture), whereas $K = 2$ showed substantial admixture in 5 of the

lions. Deleting that statement would not affect your other results, and would be more consistent with your inclusion of several values of K on the figure, which I agree are informative.

>>> We have rephrased this statement on l. 128-131 to reflect the reviewer's comment.

Lines 116-122 and Figure 2a: I think the PSMC, this text, and figure should be removed. They are irrelevant to the rest of the paper and apparently duplicate previous results in any case.

>>> We understand the reviewer's point. However, we feel it is important to keep the PSMC to put the population history of lions into a broader temporal context. Furthermore, the previous lion genomics paper from de Manuel et al. (2020), did not perform a PSMC reconstruction for Crater lions.

Line 194: The statement that neutral mutations tend to drift to fixation seems incorrect as they are no more likely to drift to fixation (i.e., frequency of 100%) than they are to extinction (frequency of 0%).

>>> We are confused by this statement because there is no mention of neutral variants in the sentence, but instead of mutations with moderate impact (i.e. non-neutral). The reviewer is correct that neutral mutations can either drift to fixation or be lost. However, the point we were trying to make was that moderate impact mutations can drift to fixation after purging of highly deleterious mutations has occurred as supported by empirical studies (e.g. Grossen et al. 2020, Nat. Comms.; Dussex et al. 2021, Cell Genomics) and simulations (Dussex et al. 2023, TREE). We have thus rephrased this sentence on l. 218-221 and now state that *'Lethal or highly deleterious alleles should be reduced in frequency relatively early during a bottleneck, while a number of moderate impact variants with a lower selection coefficient and thus less individual effect on fitness are more likely drift to fixation during a bottleneck and still contribute to inbreeding depression.'*

Lines 197-198: I think you have made a fallacious inference here. Your observation was only that Crater lions had more "High impact variants" than did another population, but that observation does not in and of itself imply anything about which population experienced a change in that quantity or in which direction. It seems at least as likely (possibly more so) that the other population lost more of its High impact variants (to purging, drift, or both), which would cause the same observed difference.

>>> The reviewer raises an interesting point. We have thus added this statement *'Alternatively, it is possible that the Serengeti and Selous lions lost more of their High impact variants through a combination of purging or drift , which would cause the same observed difference.'* on l. 226-228.

Line 200: you state that the South African individual had the lowest load, but that is not supported by figure 2b.

>>> The reviewer is most likely referring to Fig. 4b. Although not significant, the South African lion has indeed one of the lowest total load for both High and Moderate impact variants. The word ‘total’ was omitted by mistake. We have modified this statement as such to clarify this point on l. 228-231. *‘Finally, we note that the South African genome (i.e. previously identified as genetically grouping with South African lions³¹) and most likely from a zoo, shows one of the lowest total load, consistent with a scenario of purging, but high inbreeding and realised load.’*

Lines 299-301: I disagree with this statement. You simply reported stats on this population, but did not directly relate them to the life-history or behavior of lions per se.

>>> We have rephrased and toned down this statement on l. xxx, which now reads: *‘Our simulations show that rapid demographic recovery and periodic effective migration can counteract the negative genetic effects of these bottlenecks and our empirical data provide an example of how continued geographical isolation driven by habitat fragmentation may negate the positive effects of population recovery by exacerbating genetic drift.’*

Lines 303-305: This does not make sense as a general conclusion. "Demographic recovery" has a system-specific meaning. If your conclusion is that the upward trajectory of the Crater lion population is not sufficient to safeguard it against inbreeding depression, fine, but that also then implies that it has not recovered sufficiently (demographically or otherwise).

>>> We understand the reviewer’s argument and have amended this statement on l. 329-330 using ‘*increase and stability*’ instead of ‘*recovery*’.

Lines 324-328: I would delete this. It is a throwaway paragraph and not uniquely about this study.

>>> We agree with the reviewer that this paragraph is not unique to this species and have shortened this paragraph. Nevertheless, we feel that it is important to keep it to broaden our conclusion to other species/systems, especially since there have been efforts to encourage the use of simulations in conjunction with empirical data in population genomics studies (Kyriazis et al. 2023).

Line 344: please provide more information on the PCR-free library preparation protocol.

>>> we have added more details on l. 369-373.

Thank you for the opportunity to review the manuscript COMMSBIO-24-7958-T entitled “Strong male territoriality increases the risk of genome erosion in Ngorongoro Crater lions.” I enjoyed reading the paper and find the topic particularly relevant today, in our ever-changing world where wildlife populations are forced into small pockets of (potentially isolated) suitable habitat. In this particular study, the authors sequenced 15 lions from the Tanzanian Ngorongoro Crater and surrounding Serengeti to look at the effect of two subsequent population bottlenecks. The manuscript is well written and structured, so from that perspective I have little to no comments. However, the data has often been overinterpreted and there are quite a few misconceptions and inconsistencies in the manuscript. Most importantly, the authors focus their results on recent and past population declines (i.e. bottlenecks) but show no actual evidence that immigration into the Crater is reduced, nor provide any data on male territoriality, which they suggest is stronger in the Crater, thus worsening the population isolation. Therefore, the title is very misleading. The author also talks about gradual changes in genetic diversity and purging, which is impossible when you do not have generational data. The comparison of your data to other areas (Botswana and South Africa) is also misleading, consider the distance and sample size difference. My concern is that the authors have tried to oversell their results, which has been the root cause of such errors. Luckily, fifteen whole genomes is interesting enough to allow for a scientifically solid paper, but then some vigorous changes need to be applied before considering this manuscript for publication. Therefore, I recommend major revisions, which I elaborate in a point-by-point manner below.

Line 46: You state: “genetic load may not substantially change individual fitness if there is quick recovery” making it sound like some epigenetic mechanism that can change during the course of an individual’s lifetime. Consider rephrasing the sentence for clarity.

Line 48: It might be useful to explain to the reader what purging is.

Line 58: I find the “(e.g. kakapo)” quite odd here – used differently from the other brackets in the same sentence, which serve to explain a term. Not everyone may be familiar with the kakapo bird or its life history.

Line 62-64: This sentence seems entirely unrelated: critically endangered species with low reproductive rates (lions are neither)

Line 66: “due to sufficient prey abundances”; so does this sentence imply that there is suboptimal habitat and lots of prey just outside the Crater but not inside? And how do you propose that this will isolate these lions in the Crater?

Line 68: “only little immigration”. Considering that this is the central point of your study, it would be good to know what is meant by little. For instance, only six lions over the past 10 years (2 generations) would still be ten percent of the entire population, which would be significant.

Line 68: Remove “Indeed”

Line 70: I was so surprised by the statement that there is higher competition in the Crater than elsewhere (which supposedly causes reproductive isolation in the Crater) that I went to check the reference. It turns out that this source can't be verified because it's in preparation. Hence, this data won't come out before this publication. Considering the pretty extreme nature of the statement, I recommend finding other references or removing the sentence entirely. Or at least explain very well why mating opportunities are different / more limited in the Crater, when compared to other lions, and why this would isolate the population. It would also be good to explain how inbreeding avoidance, which is so common in mammals (particularly social breeders), comes into play.

Line 81: Why did it decline again the 90s?

Line 83: Rephrase sentence

Line 87: Why include genomes from Botswana and South Africa?

Line 89: Here you state "recent population decline" but the results (line 99) starts with "past population history". Beside this conflict, I don't see how these relate to the title.

Line 91: You state that your results show that reduced immigration into the Crater induced genomic erosion, but your results show absolutely no proof of that. All it does is show the effect of a population bottleneck (and founder effect)

Line 93: Biologists usually do these calculations per generation, instead of per decade, but okay; however you prefer it in this case.

Line 100: You start the results with structure analyses, but haven't clarified why you consider this necessary to answer your research question. More worryingly: you base isolation on mtDNA markers, which is completely unsuitable to look at recent population structure because it is not part of introgression and recombination (nor representative of male dispersal). Furthermore, most mitochondrial genotypes are likely to have gone lost during the bottleneck, which means that any observed difference with the surrounding area is due to drift (and not isolation).

Line 103: Structure analysis is often overinterpreted and does not properly deal with introgression. For instance, you won't ever see a case where $K=1$ is the result, so again, additional analyses are needed to conclude that Crater lions are isolated.

Line 105: This sentence is too important to write off with "however"

Line 110: As for Figure 1: the sample names "Lion01, Lion03" etc are uninformative. It'd be better to name them according to location. I can't even tell from the structure analyses which ones are Crater lions. In 1a: what are Ndulu and Endulen? And in 1c: why is there a cheetah in there? And where did South Africa go? 1d. I also can't tell from this PCA which ones are Crater lions – maybe give Crater ones a special shape?

Line 116: Again, I'm not entirely sure why you think PSMC is necessary to answer your research questions. There are three paragraphs about past climatic events, which is interesting but seemingly off-topic.

Line 132: You conclude genetic isolation again here, but isn't a bottleneck effect more likely? Figure 2b surely points that way

Line 148: Before you start to make these comparisons, it is important to remove any related individuals from the Crater (if you haven't already). You mention in the methods that you selected presumed-to-be unrelated individuals (based on field observations) but this can and should actually be tested with the genomic data.

Line 152: I agree with this statement: long ROHs and population bottlenecks are the perfect match. But once again: please reconsider the title.

Line 157: "suggesting that the parameters provide robust estimates" – why wouldn't they?

Line 161: Figure 3: You say 20 lion genomes, but in the figure I count 15. South Africa is just 1 genome, and knowing South Africa it's probably from a private reserve that stems from a reintroduction, which is likely to be mildly inbred as well. This comparison is thus pretty much pointless. What's more: the 5X sequencing has only been mentioned now and is pretty significant / concerningly low. Sequencing depth has a huge impact on measures of genetic diversity and structure, due which an ANGSD pipeline is normally recommended (based on genotype likelihoods). Otherwise just stick to the genomes that have 14X depth.

Line 181: South Africa is not outbred

Line 185: Didn't you say in the introduction that purging occurs only where there is a slow reduction in population size? The lion's bottleneck was pretty rapid – so perhaps dispersal into the Crater may actually be a more viable alternative hypothesis.

Line 186: "other populations"; please clarify which ones you mean here. I think a comparison to South Africa and Botswana are non-informative due to sample size, among other factors.

Line 190: "As theory, empirical data ..." ; if there is empirical data, it's no longer a theory You say that the early stage of a decline is characterized by such and such. But when did your sampling take place? I presume pretty recently, so not in those 'early stages', i.e. 1962

Line 195: "The patterns observed" followed by the comparison to gorilla is pretty invalid, because you actually don't have generational data: you just have one snapshot of 10 genomes. You can't say anything about a reduction of deleterious alleles through time, because you don't know what the situation was like before or after 1962.

Line 201: Like the previous comment: you definitely can't say anything about purging (which requires generational data), let alone when you look at one lion in a private park in South

Africa – or “likely” a zoo?). Also, which genome grouped with SA lions? Based on mtDNA? I don’t recall reading this.

- Line 237: So there was actually effective immigration into the Crater, twice, and you mention this only now. This seems rather important and in stark contrast to your persistent conclusion that Crater lions are isolated.
- Line 246: Again: very important and in stark contrast to the title.
- Line 250: What are Very Strongly deleterious variants? How can deleterious alleles not differ after a bottleneck, but Very Strong ones be a direct effect thereof?
- Line 252: But purging and bottlenecks don’t seem to go together, I’ve been told
- Line 301: This statement really needs to be taken out before this paper can be considered for publication. There is no scientific base for this whatsoever. In fact, it conflicts entirely with your findings, which seems to show low genetic load, likely due to effective dispersal that occurred in 1964-65 and 2013-18. These simulations are meaningless here; you can essentially just feed the programme whatever you like.
- Line 304: “Taken together, our results challenge the notion”: I disagree that your results do this at all. In the end, you have 10 genomes from the Crater, from one moment in time. Which is just fine (it can give plenty of interesting results); but it’s crucial not to oversell them.
- Line 320/27: Here, you pretty much confirm the point I wanted to make: it’s human-lion conflict and habitat fragmentation that are the problem: NOT male territoriality.
- Line 341: Here you should mention the actual genetic relatedness of your samples.
- Line 344: Here you should mention the targeted and realized sequencing depth of your genomes
- Line 372: You mention “gene flow” in the title, but only give structure-focused analyses. Structure could result from lack of gene flow, surely, but also from differences in genomic diversity. Maybe it’s worth adding EEMS, introgression and recombination rates, and genome-wide estimates of the frequency of private alleles
- Line 386: When did SA go, and why include Botswana? Including outliers like this can actually cause a polarizing effect on the structure analyses
- Line 470: What are variants of category i?
- Line 494: What is meant by genome-informed? Do you upload your WGS data?

1 Strong male territoriality increases the risk of genome erosion in 2 Ngorongoro Crater lions

Nicolas Dussex¹, Ingela Jansson², Tom van der Valk^{3,4}, Craig Packer⁵, Anita Norman²,
Bernard M. Kissui⁶, Ernest E. Mjinga⁷, Göran Spong²

1. Unit for Population Analysis and Monitoring, Swedish Museum of Natural History, SE-106 91 Stockholm, Sweden
2. Department of Wildlife, Fish, and Environmental Studies, Swedish University of Agricultural Sciences, SE-901 83 Umeå,
Sweden
3. Centre for Palaeogenetics, Svante Arrhenius väg 20C, SE-106 91 Stockholm, Sweden
4. Department of Bioinformatics and Genetics, Swedish Museum of Natural History, SE-106 91 Stockholm, Sweden
5. Department of Ecology, Evolution and Behavior, University of Minnesota, MN 55108 St. Paul, MN, USA
6. School for Field Studies, Centre for Wildlife Management Studies, Karatu, Tanzania
7. Tanzania Wildlife Research Institute (TAWIRI), Arusha, Tanzania

16 17 **Abstract**

Small, isolated populations are at greater risk of genome erosion than larger populations but
conservation efforts can lead to demographic recovery and mitigate some of the negative
genetic effects of bottlenecks. However, certain life-history traits and behaviours can contribute
to further loss of genetic variation and hamper genomic recovery. Here, we use population
genomic analyses and forward simulations to assess the genomic impacts of near extinction of
the isolated Ngorongoro Crater lion (*Panthera leo*) subpopulation. We show that 200 years of
quasi-isolation and an epizootic in 1962 resulted in a two-fold increase in inbreeding and an
excess in the frequency of highly deleterious mutations relative to Greater Serengeti lions.
However, we found little evidence for purging of genetic load through gradual increase in
inbreeding. Furthermore, forward simulations indicate that low gene flow into the Crater may
contribute to future genome erosion, with a minimum of one to five effective male migrants
29 per decade required to reduce the risk of long-term inbreeding depression and reduction in
genetic diversity. Our results suggest that in spite of a rapid demographic recovery since the
1970s, ~~strong male territoriality may reinforce the genetic isolation of the population, further~~
~~exacerbating the effects of genome erosion.~~

**Keywords:** genome erosion, bottleneck, behaviour, connectivity, migration, simulations

36 **Introduction**

Increasing anthropogenic pressure over the past centuries has had a strong impact on ecosystem
health and diversity and is often characterised by severe wildlife population declines. Such
declines have negative genetic consequences referred to as genome erosion (i.e., loss of genetic
variation, increase in genetic load, mismatch between adaptations and environment¹) which
can increase the risk of extinction by trapping species into an extinction vortex².

The rate of demographic decline and recovery as well as duration of a bottleneck will
have a strong influence on the magnitude of genome erosion^{1,3}. For instance, only a small
portion of genome-wide diversity may be lost if the decline is followed by quick demographic

rebound, whereas a sustained bottleneck will likely result in a significant loss of diversity.
Similarly, the amount of genetic load may not substantially change and impact individual
fitness if there is a quick recovery. In contrast, long-term declines characterised by gradual
increases in inbreeding may induce a purging effect, whereas rapid declines may induce an
increase in the frequency of deleterious variants³.

There has been a strong focus on examining the effect of the intensity of population
bottlenecks or founder events on genome erosion processes (e.g., ⁴⁻⁸). However, life-history
traits and breeding behaviour also play an important role in the genetic resilience to population
declines⁹, since they directly affect drift and thus the effective population size, even if a
population remains demographically stable or recovers. For instance, in polygamous species
where only few territorial males contribute genetic diversity to future generations, the genomic
recovery (e.g. reduction in inbreeding) can be relatively slow, and the random effect of genetic
drift (i.e., loss of diversity or fixation of deleterious alleles) can be particularly strong (e.g.,
kākāpō⁵). This drift effect will be even stronger if breeding events do not occur on a yearly
basis, or in species with a longer generation time and small litter size. In contrast, species with
shorter generation times and higher reproductive output are more resilient to population
fluctuations⁹ and may thus be less prone to the negative genetic effects of population declines.

A number of species with life-history traits associated with low reproductive output
(e.g. Sumatran rhinoceros, kākāpō) and characterised by geographical isolation are listed as
critically endangered (IUCN¹⁰). As a case in point, the Ngorongoro Crater (hereafter referred
to as the Crater) lion population in Tanzania, which now comprises ~60 lions (~5 prides), ~~seems~~
~~to function as an ecologically isolated population~~ due to sufficient prey abundance and
suboptimal habitat immediately outside the Crater¹¹⁻¹³. While there are no geographical
barriers to dispersal into the Crater¹³, ~~only~~ little immigration has been reported from the
adjacent Serengeti plains¹². Indeed individual identification and intensive monitoring suggest
that high competition with large local coalitions of resident males¹¹ limits the mating
opportunities of immigrant males and thus contributes to the genetic isolation of the
population¹⁴. Furthermore, consistent with an estimated 85% reduction in the range of African
lions since 1500 AD¹⁵ primarily driven by increasing anthropogenic pressure on the landscape,
the Crater population experienced several bottlenecks associated with landscape fragmentation.
Exponential human population growth and land use changes (i.e., agriculture and
pastoralism¹⁶), retaliatory and ritual lion hunts^{17,18}, colonial-era trophy hunting¹⁹ as well as
periodic rinderpest epizootics between 1890-1962 reducing ungulate populations in the Greater
Serengeti Ecosystem (GSE) exacerbated this population fragmentation over the past ~200
79 years. Moreover, the population experienced a severe decline in 1962 caused by an epizootic,
after which 15 adults and juvenile survivors founded the current population^{11,12}. Subsequently,
the population recovered but declined again in the late 90s with the number of lions close to 50
individuals over the past ~30 years. The Crater population is ~~thus~~ characterised by low diversity
and inbreeding depression with evidence for increase in levels of sperm abnormality and low
spermiogenesis^{11,12,20-22} as well as higher levels of cub mortality in the first few weeks of life¹².
~~However, to date, little is known about the consequences of population decline and isolation~~
~~on the deleterious genome-wide variation in Crater lions.~~

Here, we analyse 20 lion genomes from Tanzania (the wider Ngorongoro Conservation
Area including the Crater and the, Serengeti plains), ~~Serengeti plains~~, Botswana and South

Africa to examine the genomic consequences of the recent population decline and isolation on
 the Crater population. Genomic data and simulations show that the recent decline and reduced
 male immigration into the Crater induced an increase in genome erosion and realised load in
 the population. Furthermore, simulations reveal that a minimum of one to five effective male
 migrants per decade would be required to avoid substantial reduction in genome-wide
 variation, increases in inbreeding and genetic load.

Results and Discussion

Population structure, gene flow and past demography

We analysed 20 lion genomes to reconstruct the past population history of the Crater lions and
 to examine the consequences of isolation and recent population decline on genome-wide
 diversity and genetic load of the population. Overall, our population structure analyses
 supported genetic isolation of the Crater population with a single mtDNA clade (Fig. 1b).
 Furthermore, there was little evidence for admixture with the adjacent Serengeti plains and
 strong support (i.e. lowest cross validation error) for two genetic clusters (K=2; Fig. 1c,d).
 However, we also note that two males of unknown natal origin and one Crater-born female
 (Table S1) show some distinct ancestry.

Figure 1. Sampling, population structure and gene flow. (a) Sampling locations for African lions. Inset shows the 15 genomes sequenced for this study including 10 Ngorongoro Crater lions. **(b)** Dated mitochondrial phylogeny with divergence times (years BP). **(c)** Recent admixture graph (Serengeti = Ndotu, Endulen, Tanzania). **(d)** Admixture plot for K=2-5. **(e)** PCA for the 15 newly sequenced and five lion genomes from de Manuel *et al.*²³.

~~The demographic reconstruction based on the Pairwise Sequentially Markovian Coalescent (PSMC) showed a long term decline over the past 1Ma BP and shared demographic history~~

118 among lion populations (Fig. 2a), as previously shown by de Manuel *et al.*²³. This decline is
119 consistent with drying and cooling conditions of the first 1M of the Pleistocene, while the
120 Eemian period (c. 130,000 to 115,000 y. BP) characterised by warmer and wetter conditions
and expansion of savannas and forests showed a plateau (Fig. 2a). Subsequently this decline
continued with the next cycle of cooling²⁴ until the Holocene.

The recent past demography (i.e., ~200 generations or ~1000 y. BP²⁵) indicated a
population increase c. 1200-1000 y. BP, coinciding with a prolonged dry period in Eastern and
Equatorial Africa¹⁹, followed by a decline c. 600 y. BP, coinciding with wetter conditions of
the Little Ice Age²⁶. While speculative, this suggests that denser woodland may have been
potentially less favourable to lions (Fig. 2b).

The mitochondrial data showed that Crater lions form a monophyletic clade and that
the population diverged c. 2-9 Ka BP (Fig.1b), whereas the recent split based on nuclear data
suggests that dispersal and gene flow occurred until c. 200 y. BP (Fig. S1). The timing of this
divergence is thus consistent with the recent decline resulting from habitat fragmentation of the
past 200 y. BP and contributing to the genetic isolation of the population.

 **Figure 2. Past demography.** (a) Long-term past demography using the PSMC with a rate of 4.5×10^{-9}
 substitutions/site/generation and a generation time of 5 years²³. (b) Recent past demography inferred
 with GONE²⁵ for Crater lions over the past ~200 generations and assuming a generation time of 5
 years²³. The black line depicts the mean and shaded area the 95% CI for 10 replicates (Table S2) each
 calculated as the geometric mean over 40 independent estimates from the observed spectrum of linkage
 disequilibrium using the *Panthera* sp. recombination rate²⁷. (c) Population census size from 1962 to
 2022. Note that no data was collected between 1972-75 and 1978 and 1980 (Table S3).

Genetic diversity and load

Heterozygosity and inbreeding estimates revealed that the Crater population has the lowest
 amount of variation and is ~1.6 times as inbred compared to the larger neighbouring Serengeti
 and Selous populations ($F_{ROH-Crater} = 0.37$, $F_{ROH-Serengeti-Selous} = 0.23$; Fig. 3; Table S4).
 Furthermore, there is on average ~60% (range: 51-66%) of the F_{ROH} coefficient comprised of
 Runs of Homozygosity (ROH) ≥ 2 Mb, consistent with recent inbreeding events. When using

a recombination rate estimated in felids²⁷ and a generation time of 5 years for lions²³,
 inbreeding events characterised by ROH ≥ 2 Mb and ROH ≥ 10 Mb date to the past ~ 110 and
 ~ 30 years, respectively (Table S5). These signatures of inbreeding are thus consistent with the
 history of population decline resulting from the recent history of population fragmentation over
 the past two centuries in the region as well as with the recent decline in census size of the 2000s
 (Fig. 2c).

~~Interestingly,~~ F_{ROH} estimates for the newly-sequenced Serengeti and Selous lions in our
 dataset are consistent with previous estimates for Tanzanian lions ($F_{ROH} \approx 0.2$; Table S4)²³,
 suggesting that the parameters used here provide robust and comparable estimates of
 inbreeding.

**Figure 3. Heterozygosity and inbreeding.** (a) Heterozygosity estimates showing theta (N. het.
 sites/1000bp) for 20 lion genomes downsampled at 5X depth of coverage. Horizontal lines within
 boxplots depict the mean, bounds of boxes represent the standard deviation, and vertical bars represent
 minima and maxima. (b) Inbreeding estimates (F_{ROH}) based on the identification of Runs of
 Homozygosity (ROH) using a 100kbp sliding window for 16 lions (with depth $\geq 14X$). Complete bars
 show the proportion of genomes in ROH ≥ 100 kb (i.e., background relatedness) and lower coloured
 portions of bars show proportions in ROH ≥ 2 Mb (i.e., recent inbreeding events). Wilcoxon signed-rank
 test; * $p < 0.05$, NS. = non-significant, p-values were not adjusted for multiple comparisons.

We estimated genetic load by annotating variants in coding regions using Snpeff²⁸. When
 considering the frequency of deleterious variation among populations, the R_{xy} ratio²⁹ showed
 an excess in High impact variants and a slight deficit for Moderate impact variants in Crater
 lions relative to Serengeti and Selous lions (Fig. 4a). While the total load estimates for High
 and Moderate impact variants were not significantly different among populations, the High
 impact load was higher in the Crater population relative to other populations (Fig. 4b; Table
 S4).

Since most deleterious mutations are partially recessive and since heterozygous ones
 are hidden from selection, homozygous mutations are the most informative of negative fitness
 effects, also referred to as the realised load^{1,3}. Consistent with the higher inbreeding in the
 Crater population, we found significantly higher realised load for both High and Moderate

impact variants relative to the more outbred populations (Fig. 4c, S3). Furthermore, the
numbers of heterozygous variants for both High and Moderate categories are significantly
lower in Crater lions relative to the Serengeti population (Fig. S3), suggesting that those may
have been lost through a combination genetic drift directly associated with the recent
bottlenecks or through some early purging effect^{1,3}. Nevertheless, the higher total load relative
to other populations suggests that there has not been sufficient time for selection to purge highly
deleterious variation and that the population is most likely in the early stages of exposure to
genome erosion following the ~200 years isolation from the Serengeti population and the 1962
epizootic.

As theory, empirical data and simulations indicate, the early stage of a decline is
characterised by an increase in inbreeding and realised load and thus of inbreeding depression³.
Lethal or highly deleterious alleles should be reduced in frequency relatively early during the
bottleneck, while moderate impact variants (i.e. with a lower selection coefficient) with little
individual effect on fitness tend to drift to fixation³. The pattern observed in lions is very similar
to that of Grauer's gorilla (*Gorilla beringei graueri*), which experienced severe population
declines over the past century³⁰. However, the random effect of drift is also particularly evident
in Crater lions, with mostly High impact variants drifting to high frequencies, whereas a portion
of the Moderate impact variants may have been lost during the bottleneck. Finally, we note that
the genome previously identified as genetically grouping with South African lions²³ and most
likely coming from a zoo, shows the highest inbreeding and lowest load, consistent with a
scenario of purging³.

Figure 4. Genetic load. (a) R_{xy} ratio for high- and moderate-impact variants for Crater lions relative to Serengeti and Selous lions. $R_{xy} < 1$ or > 1 corresponds to a deficit or an excess in allele frequency, respectively, in population x (i.e., Crater, $n=10$) relative to population y (Serengeti and Selous lions, $n=5$). Whiskers represent ± 1 SD. (b) Total load estimated as the ratio of High and Moderate impact to Synonymous variants for 16 lions (with sequencing depth $\geq 14X$). (c) Realised load. Horizontal lines within boxplots depict the mean, bounds of boxes represent the standard deviation, and vertical bars represent minima and maxima. Wilcoxon signed-rank test; NS. = non-significant, p-values were not adjusted for multiple comparisons.

Since there is some evidence of inbreeding depression, for sperm abnormality and low sperm counts in Crater lions^{11,20–22}, as well as higher cub mortality in the first weeks of life¹², we examined the gene functions of genes carrying High and Moderate impact alleles (Table S6). Among genes with High impact variants (i.e., premature stop codons) at higher frequency in the Crater population relative to Serengeti and Selous lions, we found a number of genes associated with sperm morphology and male fertility (e.g., AURKC, TSHB, TLR6, CWF19L2), nervous system, (e.g., ARSG, FGD4) and immunity (e.g., TLR6; Table S7). Among genes with Moderate impact variants (i.e., Missense) at high frequency, we also found functions associated with sperm morphology and male fertility (e.g., NUTM1, CCDC87, CEP250, PRSS55, TSPYL1;) and additionally, with cardiovascular system (e.g., LAMA4,

KLK5, GAS2L3, PPP1R15B, OBSCN), nervous system (e.g., LAMA1, KCNJ16, SCN1B;
Table S7).

**Forward genome-informed simulations**

[revised manuscript text omitted]

**Population structure, gene flow and past demography**

We first performed a PCA in PLINK v2⁴⁸. We then estimated individual-based ancestry to infer
the number of genetic clusters with ADMIXTURE v1.3.0⁴⁹ for $K = 1-5$ and using the cross-
validation error estimation (--cv option). We then built a dated phylogeny using mtDNA data
for the 20 lions dataset. We first extracted bam files corresponding to the mitogenome using
the samtools view command and generated consensi sequences using ANGSD⁵⁰ with the
following parameters: -doFasta 2 -i -minQ 30 -minmapQ 30 -doCounts 1 -setMinDepth 5. We
then aligned all sequences in Geneious Prime v2022.2.2 ([https:// www. geneious.com](https://www.geneious.com))⁵¹ to
generate a multi-fasta file for the 20 lions. We then used BEAST v2⁵². We used a mutation rate
of 1.327×10^{-8} inferred from modern and ancient sequences for extinct lion species⁵³ and a
HKY model as inferred with jModelTest V2⁵⁴. We also built an admixture graph to model the
genomic relationships between the subpopulations. We first converted the filtered autosomal
SNP data to eigenstrat format and then constructed admixture graphs, iterating through all
possible graph combinations for the four lion subpopulations (Crater, Selous, Serengeti,
Botswana) using qpBrute v0.3^{55,56}. At each iteration, insertion of a new graph node was tested
against all existing branches of the graph. In cases in which a node could not be inserted without
producing f4 outliers (that is, $|Z| \geq 3$), all possible admixture combinations were also attempted.

[revised manuscript text omitted]

- 9. Capdevila, P. *et al.* Life history mediates the trade-offs among different components of
demographic resilience. *Ecol. Lett.* **25**, 1566–1579 (2022).
- 10. Ragle, J. & Remsen, D. IUCN red list of threatened species. (2010) doi:10.5281/ZENODO.46240.
- 11. Packer, C. *et al.* Case study of a population bottleneck: Lions of the Ngorongoro Crater. *Conserv.*
*Biol.* **5**, 219–230 (1991).
- 12. Packer, C. *The Lion: Behavior, Ecology, and Conservation of an Iconic Species.* (Princeton
University Press, 2023).
- 13. Jansson I., Parsons A.W., Singh N.J., Faust L., Kissui B.M., Mjingo E.E., Sandström C., Spong G.
Coexistence from a lion’s perspective: movements and habitat selection by African lions (*Panthera*
*leo*) across a multi-use landscape. *Plos One* (2024).
- 14. Jansson, I. *et al.* Genetic diversity, connectivity and dispersal patterns of a lion (*Panthera leo*)
population across a pastoralist landscape – lions in Ngorongoro Conservation Area, Tanzania. *in*
*prep.*
- 15. Morrison, J. C., Sechrest, W., Dinerstein, E., Wilcove, D. S. & Lamoreux, J. F. Persistence of large
mammal faunas as indicators of global human impacts. *J. Mammal.* **88**, 1363–1380 (2007).

[revised manuscript text omitted]

- 53. Broggini, C. *et al.* From Caves to the Savannah, the Mitogenome History of Modern Lions () and
Their Ancestors. *Int. J. Mol. Sci.* **25**, (2024).
- 54. Darriba, D., Taboada, G. L., Doallo, R. & Posada, D. jModelTest 2: more models, new heuristics
and parallel computing. *Nat. Methods* **9**, 772 (2012).
- 55. Ní Leathlobhair, M. *et al.* The evolutionary history of dogs in the Americas. *Science* **361**, 81–85
(2018).
- 56. Liu, L. *et al.* Genomic analysis on pygmy hog reveals extensive interbreeding during wild boar
expansion. *Nat. Commun.* **10**, 1992 (2019).
- 57. Li, H. & Durbin, R. Inference of human population history from individual whole-genome
sequences. *Nature* **475**, 493–496 (2011).
- 58. Terhorst, J., Kamm, J. A. & Song, Y. S. Robust and scalable inference of population history from
hundreds of unphased whole genomes. *Nat. Genet.* **49**, 303–309 (2017).
- 59. Haubold, B., Pfaffelhuber, P. & Lynch, M. mlRho - a program for estimating the population
mutation and recombination rates from shotgun-sequenced diploid genomes. *Mol. Ecol.* **19 Suppl**
**1**, 277–284 (2010).
- 60. R Core Team. *R: A Language and Environment for Statistical Computing*. (R Foundation for
Statistical Computing, Vienna, Austria, 2020).
- 61. Thompson, E. A. Identity by descent: variation in meiosis, across genomes, and in populations.
*Genetics* **194**, 301–326 (2013).
- 62. Trapnell, C. *et al.* Differential gene and transcript expression analysis of RNA-seq experiments
with TopHat and Cufflinks. *Nature Protocols* vol. 7 562–578 Preprint at
<https://doi.org/10.1038/nprot.2012.016> (2012).
- 63. Mathur, S. & DeWoody, J. A. Genetic load has potential in large populations but is realized in
small inbred populations. *Evol. Appl.* **14**, 1540–1557 (2021).
- 64. Blake, J. A. *et al.* The Mouse Genome Database: integration of and access to knowledge about the
laboratory mouse. *Nucleic Acids Res.* **42**, D810-7 (2014).
- 65. Robinson, J. A. *et al.* The critically endangered vaquita is not doomed to extinction by inbreeding
depression. *Science* **376**, 635–639 (2022).

- 66. Frankham, R. Effective population size/adult population size ratios in wildlife: a review. *Genet.*
*Res.* **89**, 491–503 (1995).
- 67. Huber, C. D., Kim, B. Y., Marsden, C. D. & Lohmueller, K. E. Determining the factors driving
selective effects of new nonsynonymous mutations. *Proc. Natl. Acad. Sci. U. S. A.* **114**, 4465–4470
(2017).
- 68. Kim, B. Y., Huber, C. D. & Lohmueller, K. E. Inference of the Distribution of Selection
Coefficients for New Nonsynonymous Mutations Using Large Samples. *Genetics* vol. 206 345–
361 Preprint at <https://doi.org/10.1534/genetics.116.197145> (2017).
- 69. Agrawal, A. F. & Whitlock, M. C. Inferences about the distribution of dominance drawn from
yeast gene knockout data. *Genetics* **187**, 553–566 (2011).
- 70. Huber, C. D., Durvasula, A., Hancock, A. M. & Lohmueller, K. E. Gene expression drives the
evolution of dominance. *Nat. Commun.* **9**, 2750 (2018).

(d)

(b)

(e)

(c)

(a)**(b)****(c)**
(a)**(b)**

(a)**(b)****(c)****(d)**
Reviewers' comments:

Reviewer #1 (Remarks to the Author):

Thank you for the opportunity to review the manuscript COMMSBIO-24-7958-T entitled “Strong male territoriality increases the risk of genome erosion in Ngorongoro Crater lions.” I enjoyed reading the paper and find the topic particularly relevant today, in our ever-changing world where wildlife populations are forced into small pockets of (potentially isolated) suitable habitat. In this particular study, the authors sequenced 15 lions from the Tanzanian Ngorongoro Crater and surrounding Serengeti to look at the effect of two subsequent population bottlenecks. The manuscript is well written and structured, so from that perspective I have little to no comments. However, the **data has often been overinterpreted and there are quite a few misconceptions and inconsistencies in the manuscript**. Most importantly, the authors focus their results on recent and past population declines (i.e. bottlenecks) but show no actual evidence that immigration into the Crater is reduced, nor provide any data on male territoriality, which they suggest is stronger in the Crater, thus worsening the population isolation. Therefore, the **title is very misleading**. The author also talks about gradual changes in genetic diversity and purging, **which is impossible when you do not have generational data**. The comparison of your data to other areas (Botswana and South Africa) is also misleading, consider the distance and sample size difference. My concern is that the authors have tried to oversell their results, which has been the root cause of such errors. Luckily, fifteen whole genomes is interesting enough to allow for a scientifically solid paper, but then some vigorous changes need to be applied before considering this manuscript for publication. Therefore, I recommend major revisions, which I elaborate in a point-by-point manner below.

>>> We thank the reviewer for their constructive comments, which have allowed us to greatly improve the manuscript. We have toned down our interpretation of the results and reframed our conclusion around the impact of bottleneck and habitat-fragmentation on genome erosion in Crater lions. Nevertheless, we have now added more details on the evidence supporting higher male territoriality in the Crater in the introduction (Packer 2023, Chapter 10). While only speculative, we believe it is worth mentioning the potential role of male territoriality in the pattern of genome erosion observed.

However, we respectfully disagree that only generational (i.e. temporally-spaced) data allows to show evidence of purging. While temporal data is ideal to test for purging ‘in real time’ as we have done previously (see Dussex et al. 2021, 2023; von Seth et al. 2021, 2022), a number of studies have used comparative analyses with modern data alone to test for evidence of purging (e.g. Ibex, Grossen et al. 2020; rattlesnake, Mathur et al. 2023; Montezuma quail, Mathur & deWoody 2021; Svalbard reindeer, Dussex et al. 2023; Moose, Kyriazis et al. 2023; Orkney voles, Wang et al. 2023). Furthermore, simulations are a valuable alternative and addition to temporally-spaced data as they allow to validate the

interpretation (in this case, lack of purging in Crater lions) of empirical data (see Kyriazis et al. 2023, *American Naturalist*, for a review). Importantly, the parameters used in these models are not approximations but have for most of them been measured in the field and over several decades by co-authors of this manuscript who are lion specialists (Ingela Jansson, Craig Packer).

Finally, we agree that the sample size and geographical representation is patchy and can limit the scope of the study. However, as shown in similar studies (e.g. de Manuel et al. 2020), this can provide informative points of comparison and clues on past demography for instance (e.g. using Runs of Homozygosity). We thus feel that they should not be removed from the manuscript altogether. Furthermore, most of the results and discussion focuses on the populations with the largest sample sizes (i.e., Crater and Greater Serengeti).

<<< I appreciate the changes and believe the manuscript has improved substantially. However, there are still some issues that need attention. I will elaborate on these below.

Line 46: You state: “genetic load may not substantially change individual fitness if there is quick recovery” making it sounds like some epigenetic mechanism that can change during the course of an individual’s lifetime. Consider rephrasing the sentence for clarity.

>>> We have rephrased this sentence on l. 49-50 which now reads: *‘Similarly, the amount of genetic load may not substantially change and thus impact individual fitness in future generations if there is a quick population recovery.’*

<<< Done

Line 48: It might be useful to explain to the reader what purging is.

>>> We have added a brief explanation of purging on l. 50-53 and state that *‘In contrast, long-term declines characterised by gradual increases in inbreeding may induce a reduction in deleterious variation through purifying selection, in a process referred to as purging, whereas rapid declines may induce an increase in the frequency of deleterious variants’*.

<<< Done

Line 58: I find the “(e.g. kakapo) quite odd here – used differently from the other brackets in the same sentence, which serve to explain a term. Not everyone may be familiar with the kakapo bird or its life history.

>>> We have clarified this statement on l. 64.

<<< Done

Line 62-64: This sentence seems entirely unrelated: critically endangered species with low reproductive rates (lions are neither)

>>> We agree with the reviewer and have removed this sentence.

<<< Done

Line 66: “due to sufficient prey abundances”; so does this sentence imply that there is suboptimal habitat and lots of prey just outside the Crater but not inside? And how do you propose that this will isolate this lions in the Crater?

>>> Yes, this is what we meant. This sentence suggests that the limited opportunities for pride occupancy due to suboptimal habitat and resources outside the Crater may be contribute the population’s isolation. We have now clarified and rephrased this sentence on l. 67-71: ‘...seems to function as an ecologically isolated population. The Crater offers high prey abundance and minimal human threats whereas the surrounding landscape is a multi-use area shared with pastoralists and their livestock, where natural prey densities are lower and more variable, and where human-lion conflicts are frequent. Thus, opportunities for pride establishment outside the Crater are limited¹²⁻¹⁴.’

<<< **Done**

Line 68: “only little immigration”. Considering that this is the central point of your study, it would be good to know what is meant by little. For instance, only six lions over the past 10 years (2 generations) would still be ten percent of the entire population, which would be significant.

>>> This is a fair point. We now clarify this statement on l. 72-73.

<<< **Done**

Line 68: Remove “Indeed”

>>> We have removed ‘indeed’.

<<< **Done**

Line 70: I was so surprised by the statement that there is higher competition in the Crater than elsewhere (which supposedly causes reproductive isolation in the Crater) that I went to check the reference. It turns out that this source can’t be verified because it’s in preparation. Hence, this data won’t come out before this publication. Considering the pretty extreme nature of the statement, I recommend finding other references or removing the sentence entirely. Or at least explain very well why mating opportunities are different / more limited in the Crater, when compared to other lions, and why this would isolate the population. It would also be good to explain how inbreeding avoidance, which is so common in mammals (particularly social breeders), comes into play.

>>> We have now removed the reference to the ‘in prep.’ manuscript and have expanded on the territoriality argument based on Chapter 10 of ‘Packer, C. The Lion: Behavior, Ecology, and Conservation of an Iconic Species. (Princeton University Press, 2023).’ We now provide more details supporting the argument of territoriality in male Crater lions and now only suggest that it could also contribute to the genetic isolation of the population.

We have also added a sentence on inbreeding avoidance on l. 81-83.

<<< I am happy with the changes. However, it must be kept in mind that male territorially is not necessarily based on resource limitations / competition for food, but mainly to monopolize mating opportunities / competition for females. And I still severely doubt that lions outside the Crater would give up their ladies more easily because there is less food available. Hence, I don't think mating opportunities are necessarily reduced in the Crater compared to what can be considered natural (males will defend themselves no matter what), but more that it is increased outside the Crater (due to the removal of dominant males through human-wildlife conflict). And if bachelor lions move out of the Crater looking for mates, you can't technically speak of genetic isolation – although, of course, it may enhance genomic erosion within the Crater, as you mentioned. With this in mind, line 82 sounds a bit contradictory: “In spite of isolation [...] and long-distance male dispersal”. I would change the sentence to: “This may be counteracted by inbreeding avoidance, as it has been found that mating among related pride members are rare (16,17), and occasional long-distance male dispersal.”

Line 81: Why did it decline again the 90s?

>>> We have now added some information on l. 94-96. The sentence now reads: *‘After a period of recovery, the population declined again in the late 90s although the real cause is unknown²³ and in 2001, from a combination of tick-borne disease and canine distemper virus (CDV)²⁴, with the number of lions close to 50 individuals over the past ~30 years’.*

<<< Done

Line 83: Rephrase sentence

>>> We have rephrased and split this sentence in two sentences on l. 99-100.

<<< Done

Line 87: Why include genomes from Botswana and South Africa?

>>> We used those genomes, as well as two other published genomes from Tanzania as a point of comparison and to increase the scope of our population structure analyses. We have now clarified this sentence on l. 104-110.

<<< Done

Line 89: Here you state “recent population decline” but the results (line 99) starts with “past population history”. Beside this conflict, I don't see how these relate to the title.

>>> We have now rephrased the title. We have also removed ‘past’ on l. 122 to remove this confusion.

<<< Done

Line 91: You state that your results show that reduced immigration into the Crater induced genomic erosion, but your results show absolutely no proof of that. All it does is show the effect of a population bottleneck (and founder effect)

>>> We have rephrased this sentence on l. 111-113 and now state that '*Genomic data suggest that the recent decline induced an increase in genome erosion and realised load in the Crater population while simulations suggest that reduced male immigration could have potentially contributed to the observed pattern.*'

<<< **Done**

Line 93: Biologists usually do these calculations per generation, instead of per decade, but okay; however you prefer it in this case.

>>> We take note of this comment.

<<< **Done**

Line 100: You start the results with structure analyses, but haven't clarified why you consider this necessary to answer your research question. More worryingly: you base isolation on mtDNA markers, which is completely unsuitable to look at recent population structure because it is not part of introgression and recombination (nor representative of male dispersal). Furthermore, most mitochondrial genotypes are likely to have gone lost during the bottleneck, which means that any observed difference with the surrounding area is due to drift (and not isolation).

>>> We have now added a justification for performing population structure analyses on l. 125-127.

>>> The reviewer raises a fair point regarding the mitochondrial phylogeny and we have now removed the mitochondrial phylogeny. However, we do not base the isolation solely on this analysis and provide an admixture plot and PCA and also added an estimation of the divergence time between Ngorongoro Crater and Serengeti plains genomes.

<<< **Done**

Line 103: Structure analysis is often overinterpreted and does not properly deal with introgression. For instance, you won't ever see a case where $K=1$ is the result, so again, additional analyses are needed to conclude that Crater lions are isolated.

>>> We have clarified the presentation of the admixture results and have provided the Cross Validation values, as requested by R2's comments on l. 128-131.

<<< **Done**

Line 105: This sentence is too important to write off with "however"

>>> We have removed 'however' on l. 131.

<<< **Done**

Line 110: As for Figure 1: the sample names “Lion01, Lion03” etc are uninformative. It’d be better to name them according to location. I can’t even tell from the structure analyses which ones are Crater lions. In 1a: what are Ndulu and Endulen? And in 1c: why is there a cheetah in there? And where did South Africa go? 1d. I also can’t tell from this PCA which ones are Crater lions – maybe give Crater ones a special shape?

>>> We have relabelled lion genomes in the admixture plot (now Fig. 1c) and removed the mitochondrial tree. In the legend of the PCA (now Fig. 1b), we have clarified the legend, indicating that Endulen, Ndutu and Tanzania are grouped into Greater Serengeti. We have also removed the admixture graph. We also note that Cheetah was there as an outgroup.

<<< **Done**

Line 116: Again, I’m not entirely sure why you think PSMC is necessary to answer your research questions. There are three paragraphs about past climatic events, which is interesting but seemingly off-topic.

>>> We understand the reviewer’s point. However, we feel it is important to keep the PSMC to put the population history of lions into a broader temporal context. Furthermore, the previous lion genomics paper from de Manuel et al. (2020), did not perform a PSMC reconstruction for Crater lions.

<<< **I agree that it is interesting and useful in this paper, but perhaps it would be worth adding it as a research aim, so that it seems less “off-topic”**

Line 132: You conclude genetic isolation again here, but isn’t a bottleneck effect more likely? Figure 2b surely points that way

>>> What we mean is that the main drivers of this isolation is the bottleneck and that habitat fragmentation outside the Crater may exacerbate this genetic isolation. We have rephrased this sentence on l. 134-137. such as: *‘The timing of this divergence is thus consistent with the recent bottleneck and increase in habitat fragmentation over the past 200 y. BP which both contribute to the genetic isolation of the population.’*

<<< **Done**

Line 148: Before you start to make these comparisons, it is important to remove any related individuals from the Crater (if you haven’t already). You mention in the methods that you selected presumed-to-be unrelated individuals (based on field observations) but this can and should actually be tested with the genomic data.

>>> We have estimated relatedness and mention it in the methods on l. 365-367.

<<< **Done**

Line 152: I agree with this statement: long ROHs and population bottlenecks are the perfect match. But once again: please reconsider the title.

>>> We thank the reviewer for this comment.

<<< I noticed that the title has changed; there is just a slight error in it: increases should be increase: “Constraints to gene flow increase the risk of genome erosion in Ngorongoro Craters lions”

Line 157: “suggesting that the parameters provide robust estimates” – why wouldn’t they?

>>> We have removed this statement.

<<< Done

Line 161: Figure 3: You say 20 lion genomes, but in the figure I count 15.

>>> We have corrected this and refer to 16 lions in Figure 3.

<<< Done

South Africa is just 1 genome, and knowing South Africa it’s probably from a private reserve that stems from a reintroduction, which is likely to be mildly inbred as well. This comparison is thus pretty much pointless.

>>> We respectfully disagree. Even though a sample size of one is not statistically powerful, even a single genome can provide relevant information. For instance, ROH distribution can inform about the past population history of a population by allowing to estimate the timing of inbreeding events. Thus, we feel that having this point of comparison is still informative.

<<< Done

What’s more: the 5X sequencing has only been mentioned now and is pretty significant / concerningly low. Sequencing depth has a huge impact on measures of genetic diversity and structure, due which an ANGSD pipeline is normally recommended (based on genotype likelihoods). Otherwise just stick to the genomes that have 14X depth.

>>> This is an error on our part. We were referring to 16 high coverage genomes and we did not downsample to 4X but to 14X. This has been amended on l. 187, 396 and 436.

<<< Done

Line 181: South Africa is not outbred

>>> The reviewer is correct. We have now added a reference to the Serengeti and Selous populations on l. 206.

<<< Done

Line 185: Didn’t you say in the introduction that purging occurs only where there is a slow reduction in population size? The lion’s bottleneck was pretty rapid – so perhaps dispersal into the Crater may actually be a more viable alternative hypothesis.

>>> In the introduction, we state that purging is facilitated when there is a gradual decline. However, some purging will occur anyways, even in the early stages of a decline and assuming that there is some increase in inbreeding, although a longer period of decline and

longer exposure of deleterious alleles in homozygous state will facilitate this. In fact, purging (i.e. purifying selection) happens all the time, but the speed and efficiency of this process will depend on population size (i.e. N_e) and selection coefficient of mutations, as we discussed in Dussex et al. (2023, TREE). Furthermore, effective dispersal will not cause a purging effect, but instead the opposite through the introduction of new deleterious alleles (e.g. Isle Royale wolf, Robinson et al. 2019, Science; Dussex et al. 2023, TREE).

<<< **Done**

Line 186: “other populations”; please clarify which ones you mean here. I think a comparison to South Africa and Botswana are non-informative due to sample size, among other factors.

>>> Botswana is not included in this analysis and we only compare Crater to Serengeti, Selous and South Africa. We have clarified this statement on l. 211.

<<< **Done**

Line 190: “As theory, empirical data ...” ; if there is empirical data, it’s no longer a theory. You say that the early stage of a decline is characterized by such and such. But when did your sampling take place? I presume pretty recently, so not in those ‘early stages’, i.e. 1962

>>> We make a general statement and refer here to three elements: theory, empirical data and simulations. This statement is based on a number of empirical studies and key reviews on the topic (e.g. Hedrick & Garcia-Dorado 2016; Dussex et al. 2023). The sampling date of our samples is not relevant to this statement.

<<< **Done**

Line 195: “The patterns observed” followed by the comparison to gorilla is pretty invalid, because you actually don’t have generational data: you just have one snapshot of 10 genomes. You can’t say anything about a reduction of deleterious alleles through time, because you don’t know what the situation was like before or after 1962.

>>> We respectfully disagree with this comment. While temporal data is ideal to test for purging as we have done in the past (see Dussex et al. 2021, von Seth et al. 2021, 2022), a growing number of studies have used comparative analyses with modern data alone to test for evidence of purging (e.g. Ibex, Grossen et al. 2020; rattlesnake, Mathur et al. 2023; Montezuma quail, Mathur & deWoody 2021).

<<< **I am happy with the changes (aside from a small error in line 220: are more likely to drift). That being said: you can say something about the presence of deleterious alleles and compare them to a large, randomly breeding population, but not directly conclude purging has occurred within Ngorongoro through the generations. But okay, I don’t think we are going to agree on this matter ;) Side notes:**

Mathur & Woody, 2021, do exactly that: “The smaller and more isolated Texas population is significantly more inbred than the large Arizona and the intermediate-sized New Mexico populations we surveyed. [article] Thus, our genomic data illustrate and quantify the

incidence of potential genetic load in large populations (Arizona) relative to the realized genetic load in small, inbred populations such as Texas.”

Grossen et al. 2020 have samples from ancestral and derived populations so they can rightfully say something about purging: “Accumulation and purging of deleterious mutations. Consistent with the fact that all extant Alpine ibex originate from the Gran Paradiso, ..”

Mathur et al. 2023 does not conclude purging based on their results. They only say “These empirical findings may result from the fact that repeated historical bottlenecks can lead to purging of deleterious mutations from small populations, leading to reduced levels of genetic load (24-26).” in the introduction.

Line 201: Like the previous comment: you definitely can’t say anything about purging (which requires generational data), let alone when you look at one lion in a private park in South Africa – or “likely” a zoo?). Also, which genome grouped with SA lions? Based on mtDNA? I don’t recall reading this.

>>> We respectfully disagree with this comment as well. We cite de Manuel et al. (2020, PNAS) to support the statement that this genome is most likely from South Africa. We have clarified this statement on l. 228-231.

<<< **Done(I meant to say that you can’t use a zoo lion from SA to conclude that there is purging in Ngorongoro).**

Line 237: So there was actually effective immigration into the Crater, twice, and you mention this only now. This seems rather important and in stark contrast to your persistent conclusion that Crater lions are isolated.

>>> We now mention these immigration events in the introduction on l. 97-98.

<<< **Done**

Line 246: Again: very important and in stark contrast to the title.

>>> We now mention these immigration events in the introduction on l. 97-98.

<<< **Done**

Line 250: What are Very Strongly deleterious variants? How can deleterious alleles not differ after a bottleneck, but Very Strong ones be a direct effect thereof?

>>> The explanation is given in the following sentence in the text on l. 282. Variants with the highest selection coefficient and thus highest impact on fitness will be purged more quickly, early and efficiently during the bottleneck, whereas those with lower selection coefficient are more likely to get fixed by drift over the long term due to their lower impact on fitness (see Dussex et al. 2023, TREE for a review on the dynamics of genetic load).

<<< **Done**

Line 252: But purging and bottlenecks don't seem to go together, I've been told

>>> We are not sure we understand what the reviewer is referring to here. In fact purging is an ongoing process whose effectiveness will vary in function of the effective population size, inbreeding, strength of bottleneck and the selection coefficients of deleterious mutations. It is true that during a sudden and severe bottleneck, drift will be strong and may tend to lead to an increase in frequency of deleterious alleles. However, the most deleterious mutations will be purged more quickly or in the earliest stage of the bottleneck as inbreeding increases. This point is illustrated in Fig. 3 of Dussex et al. (2023, TREE) We have clarified this statement on l. 282. and state that *'This is not entirely surprising since the most deleterious variants should be purged relatively early during a bottleneck through purifying selection as they have the most impact on fitness'*.

<<< **Done**

Line 301: This statement really needs to be taken out before this paper can be considered for publication. There is no scientific base for this whatsoever. In fact, it conflicts entirely with your findings, which seems to show low genetic load, likely due to effective dispersal that occurred in 1964-65 and 2013-18. These simulations are meaningless here; you can essentially just feed the programme whatever you like.

>>> We have amended this statement on l. 326-327 in light of R1 and R2's comments. However, we respectfully disagree with the reviewer on this point. First, as stated above, we have provided more information of high male territoriality in the Crater, in the introduction on l. 67-83. Secondly, the Crater population has a higher total and realised load compared to the other populations (Fig. 4). Importantly, key studies and reviews have shown that effective dispersal into an inbred population actually increases genetic load and its expression (see Robinson et al. 2019, Science; Dussex et al. 2023, TREE). Even if a population is not inbred, gene flow from a distinct population is more likely to increase total load than reduce it. What is also crucial, is not only the amount of load (correlated to N_e) but the distribution of allele frequencies of deleterious variants. Third, we disagree that simulations are meaningless. In fact, there has been repeated calls to use simulations in population genomics studies (e.g. Haller & Messer, American Naturalist, 2023; Kyriazis et al. American Naturalist, 2023) as well as many empirical studies (e.g. Kyriazis et al. 2023, MBE; Dussex et al. 2021, 2023; Speak et al. 2024, MolEcol; Dussex et al. 2024, SEPA report) to assess the interpretation of results or predict future genomic trajectories of populations. Importantly, the parameters used for our simulations were informed by ecological field studies by Craig Packer, Ingela Jansson, and Göran Spong in the Crater, Serengeti and Selous.

<<< **Done.**

Like 304: "Taken together, our results challenge the notion": I disagree that your results do this at all. In the end, you have 10 genomes from the Crater, from one moment in time. Which is just fine (it can give plenty of interesting results); but it's crucial not to oversell them.

>>> We have also rephrased this statement on l. 329-331.

<<< **Done**

Line 320/27: Here, you pretty much confirm the point I wanted to make: it's human-lion conflict and habitat fragmentation that are the problem: NOT male territoriality.

>>> We have toned down our interpretation throughout the manuscript.

<<< **Done**

Line 341: Here you should mention the actual genetic relatedness of your samples.

>>> We have estimated pairwise relatedness and mention it on l. 365-367.

<<< **Done**

Line 344: Here you should mention the targeted and realized sequencing depth of your genomes

>>> We have added this information on l. 372.

<<< **Done**

Line 372: You mention "gene flow" in the title, but only give structure-focused analyses. Structure could result from lack of gene flow, surely, but also from differences in genomic diversity. Maybe it's worth adding EEMS, introgression and recombination rates, and genome-wide estimates of the frequency of private alleles

>>> We have now removed the mention of gene flow in this title since we are only referring to population structure and isolation.

<<< **Done**

Line 386: When did SA go, and why include Botswana? Including outliers like this can actually cause a polarizing effect on the structure analyses

>>> This was an omission on our part. However, as requested by the editor, we have now completely removed the Admixture graph analysis.

<<< **Done**

Line 470: What are variants of category i?

>>> This symbol refers to any category considered. We have now italicised the symbol for clarification.

<<< **Done**

Line 494: What is meant by genome-informed? Do you upload your WGS data?

>>> No, we have not used genetic data in the simulations. We have now replaced '*genome-informed*' with '*genomic*' for clarity.

<<< Done

Other comments:

- Line 25: 'Furthermore' probably fits better than 'However'. The fact that you see inbreeding depression sort of implies that purging did not take place, supporting your results. I would then remove the next 'Furthermore'
- Line 40: is = are (anthropogenic activities ... are)
- Line 57: Perhaps remove the second 'even if'
- Line 73: here you state 'only one breeding male establishing in the Crater between 1965 and 2013', but in line 97 it is several.
- Line 58: Add a comma after inbreeding
- Line 94: "although the real cause is unknown" should be between commas (maybe change to "for unknown causes")
- Line 133: Remove comma
- Line 176: Comprised of = comprising
- Line 179: This is funny sentence: ("history of population decline resulting from the recent history of population fragmentation [...] as well as the recent decline in census size"). Maybe just say: These signatures of inbreeding are thus consistent with the recent history of population decline resulting from habitat fragmentation over the past two centuries in the region.
- Line 328: theerby = thereby
- Line 337: reduce = reducing, increase = increasing

1 Constraints to gene flow increases the risk of genome erosion in 2 Ngorongoro Crater lions

[revised manuscript text omitted]

(a)**(b)****(c)**
(a)**(b)**

(a)**(b)****(c)****(d)**